# Apoplastic class III peroxidases PRX62 and PRX69 promote Arabidopsis root hair growth at low temperature

Javier Martínez Pacheco [1], Philippe Ranocha[2], Luciana Kasulin[1], Corina M. Fusari [3], Lucas Servi [4], Ariel. A. Aptekmann [5], Victoria Berdion Gabarain[1], Juan Manuel Peralta [1], Cecilia Borassi[1], Eliana Marzol[1], Diana Rosa Rodríguez-Garcia[1], Yossmayer del Carmen Rondón Guerrero[1], Mariana Carignani Sardoy [1], Lucía Ferrero[6], Javier F. Botto [7], Claudio Meneses[8], Federico Ariel[6], Alejandro D. Nadra [5], Ezequiel Petrillo [4], Christophe Dunand[2] & José M. Estevez [1,8,9✉]

Root Hairs (RHs) growth is influenced by endogenous and by external environmental signals that coordinately regulate its final cell size. We have recently determined that RH growth was unexpectedly boosted when *Arabidopsis thaliana* seedlings are cultivated at low temperatures. It was proposed that RH growth plasticity in response to low temperature was linked to a reduced nutrient availability in the media. Here, we explore the molecular basis of this RH growth response by using a Genome Wide Association Study (GWAS) approach using *Arabidopsis thaliana* natural accessions. We identify the poorly characterized PEROXIDASE 62 (PRX62) and a related protein PRX69 as key proteins under moderate low temperature stress. Strikingly, a cell wall protein extensin (EXT) reporter reveals the effect of peroxidase activity on EXT cell wall association at 10 °C in the RH apical zone. Collectively, our results indicate that PRX62, and to a lesser extent PRX69, are key apoplastic PRXs that modulate ROS-homeostasis and cell wall EXT-insolubilization linked to RH elongation at low temperature.

[1] Fundación Instituto Leloir and IIBBA-CONICET, Av. Patricias Argentinas 435, Buenos Aires C1405BWE, Argentina. [2] Université de Toulouse, UPS, UMR 5546, Laboratoire de Recherche en Sciences Végétales, Université de Toulouse, CNRS, UPS, Toulouse INP, Castanet-Tolosan 31326, France. [3] Centro de Estudios Fotosintéticos y Bioquímicos, Universidad Nacional de Rosario, 2000, Rosario, Santa Fe, Argentina. [4] Instituto de Fisiología, Biología Molecular y Neurociencias (IFIBYNE-UBA-CONICET) and Facultad de Ciencias Exactas y Naturales, Universidad de Buenos Aires, Ciudad Universitaria, Buenos Aires, Argentina. [5] Departamento de Fisiología, Biología Molecular y Celular, Instituto de Biociencias, Biotecnología y Biología Traslacional (iB3), Facultad de Ciencias Exactas y Naturales, Universidad de Buenos Aires, Ciudad Universitaria, Buenos Aires C1428EGA, Argentina. [6] Instituto de Agrobiotecnología del Litoral, Universidad Nacional del Litoral, CONICET, FBCB, Centro Científico Tecnológico CONICET Santa Fe, Colectora Ruta Nacional No 168km. 0, Paraje El Pozo, Santa Fe 3000, Argentina. [7] IFEVA, UBA, CONICET, Facultad de Agronomía, Universidad de Buenos Aires, C1417DSE Ciudad Autónoma de Buenos Aires, Buenos Aires, Argentina. [8] Centro de Biotecnología Vegetal, Facultad de Ciencias de la Vida, Universidad Andres Bello, Santiago, Chile. [9] ANID - Millennium Science Initiative Program - Millennium Institute for Integrative Biology (iBio) and Millennium Nucleus for the Development of Super Adaptable Plants (MN-SAP), Santiago, Chile. ✉email: jestevez@leloir.org.ar

Root hairs (RH) have emerged as an excellent model system for studying cell size regulation since they can elongate several hundred-fold their original dimensions. The rate at which cells grow is determined both by cell-intrinsic factors as well as by external environment signals. RHs represent an important proportion of the surface root area, crucial for nutrient uptake and water absorption. RH growth is controlled by the interaction of several proteins, including the bHLH transcription factor (TF) RSL4 (*ROOT HAIR DEFECTIVE 6-LIKE 4*), which defines the final RH length[1,2] as well as the related TF RSL2 (*ROOT HAIR DEFECTIVE 6 SIX-LIKE 2*)[3,4]. Together with the developmental and genetic pathways, several hormones are important modulators of RH cell growth[2,5–7]. In addition, abnormal Reactive Oxygen Species (ROS) accumulation in RHs triggers either exacerbated growth or cell bursting. Exogenous $H_2O_2$ inhibited RH polar expansion, while treatment with ROS scavengers (e.g., ascorbic acid) caused RH bursting[8], reinforcing the notion that a balanced ROS-homeostasis is required to modulate cell elongation by affecting cell wall properties. Accordingly, apoplastic ROS ($_{apo}$ROS) produced in the apoplast (specifically $_{apo}H_2O_2$) coupled to apoplastic Class III peroxidase (PRX) activity directly affects the degree of cell wall crosslinking[9] by oxidizing cell wall compounds and leading to the stiffening of the wall in peroxidative cycles (PC)[8]. In addition, $_{apo}$ROS coupled to PRX activity enhances non-enzymatic wall loosening by producing oxygen radical species (e.g.$^{•}$OH) and promoting polar-growth in hydroxylic cycles (HC)[10]. Finally, PRXs also contribute to the production of superoxide radical ($O_2^{•-}$) pool together with NADPH oxidase/respiratory burst oxidase homolog (RBOH) proteins by oxidizing singlet oxygen in the oxidative cycle (OC), thereby affecting $_{apo}H_2O_2$ levels. Given their multiple enzymatic activities in vivo, apoplastic PRXs emerge as versatile regulators of rapid cell elongation. Assigning specific functions to each of the numerous PRXs (75 encoded in Arabidopsis;[11] and even more in other plant types, e.g., 138 encoded in Rice[12]) has been challenging. Recently, three PRXs possibly linked to Tyr-crosslinking of cell wall extensins (EXTs), PRX01, PRX44, and PRX73, were characterized as important regulators of RH growth under low-nutrient conditions[13]. These RH-specific PRXs are under the direct control of the TF RSL4, a master regulator of RH cell size[1,2,14]. In addition, other PRXs were postulated to crosslink EXTs in aerial plant tissues. PRX09 and PRX40 were proposed to crosslink EXTs during tapetum development, and both, PRXs were able to crosslink EXT23 in transient expression experiments[15].

Although there is a fairly well-known mechanistic view of how RH cell expands, the environmental signals that trigger the cell elongation process remain currently unknown. Due to its important role in root physiology, it has been anticipated that RH would be highly susceptible to environmental stresses such as heat or moderate temperature increase, which trigger extensive DNA methylation, transcriptomic and proteomic changes[16–18]. Although RH development during cold acclimation remains largely unexplored, it has been observed that many RH-related genes respond to cold in the whole plant or seedlings[19–21]. It is known that plants may perceive cold by a putative receptor at the cell membrane and initiate a signal to activate the cold-responsive genes and transcription factors for mediating stress tolerance[22–26]. Previously, we have shown that the plant long noncoding RNA (lncRNA) *AUXIN REGULATED PROMOTER LOOP* (*APOLO*) recognizes the locus encoding the RH (RH) master regulator RHD6 (*ROOT HAIR DEFECTIVE 6*) and controls *RHD6* transcriptional activity leading to cold-enhanced RH elongation through the consequent activation of *RSL4*[27]. In addition, *APOLO* is able to bind and positively regulate the expression of several cell wall EXTENSIN (EXT) encoding genes,

including *EXT3*, a key regulator for RH growth[28]. Unexpectedly, our previous results indicate that the low temperatures (10 °C) are able to trigger an exacerbated RH growth compared with cell expansion at room temperature[27,28]. To explore the molecular basis of this strong growth response, we conducted Genome Wide Association Studies (GWAS) on *Arabidopsis thaliana* natural accessions and identified the uncharacterized PEROXIDASE 62 (PRX62) as a key protein that regulates the conditional growth under a moderate low temperature stress. In addition, we also identified a second PRX, i.e., PRX69, as an important player in this developmental response. Both PRX62 and PRX69 are key enzymes to trigger RH growth, likely by participating in a ROS-mediated mechanism of polar cell growth at low temperatures. The expression of both PRX encoding genes could be under the regulation of RSL4, which has a direct binding to *PRX69* promoter-specific regions. Transcriptomic analyses revealed that upon *PRX62* and *PRX69* knockout, several other PRXs and cell wall EXTs encoding genes were differentially expressed, hinting at a compensatory mechanism.

## Results

**PRX62 and *PRX69* emerged as positive regulators of RH growth at low temperatures**. In order to identify the natural genetic components involved in RH growth under low temperature conditions (at 10 °C), we analyzed 108 natural *A. thaliana* accessions originated from contrasting environments (Europe, Asia, Africa, and North America, Supplementary Fig. 1). We assessed RH growth for each seedling accession grown under 22 °C for 5 days, and then transferred them to 10 °C for 3 days. RH length was the phenotypic trait recorded for each accession, and compared to seedlings grown at 22 °C for 8 days, taken as a control. We observed 15-folds range of natural variation for average RH length (148–2218 μm) in the accessions grown at 10 °C (Supplementary Fig. 2a; Supplementary Data 1) in contrast with a lower variability (~7-folds) and significantly shorter overall RH cells when seedlings were grown at 22 °C (136–1034 μm). There is a strong positive correlation ($R^2 = 0.981$) for RH length from accessions grown at 22 °C→10 °C compared to plants growth at 22 °C (Supplementary Fig. 2b), indicating that accessions respond in the same manner to a temperature decrease but varying in intensity. Only the most contrasting accession are shown as examples (Fig. 1a). Thus, moderate low temperature triggers RH polar-growth across *Arabidopsis* ecotypes by a yet unknown molecular mechanism. To identify candidate genes involved in RH-growth response at moderate low temperature, we performed GWAS (GWAPP web tool[29]) using as input data the RH length recorded only at 22 °C or at 22 °C →10 °C for 107 accession with genotypic data available (Supplementary Data 1). GWAS was performed for both conditions using SNPs with a minor allele frequency ≥ 10% to avoid the identification of very rare alleles. GWAS for RH length from plants grown at 22 °C did not give any significant associations (Fig. 1b). On the other hand, for plants grown at 22 °C→10 °C, GWAS showed a significant association for RH length on Chromosome 5, at SNP m190905 (TAIR10 position 15847854; LOD [for log of the odds] = 6.01, FDR = 0.06). When we looked again at GWAS from RH length at 22 °C, we identified a non-significant peak in the Manhattan plot co-localizing with SNP m190905. This evidences both that mild low temperature triggers a higher variation in RH length due to polymorphisms on Chromosome 5 and that there is a residual variation in RH length when plants grown at 22 °C, though not enough to be significantly detected. Based on this, we cannot exclude that PRX62 may be still playing a relevant role in RH growth at 22 °C.

The SNP m190905 is located in the intron of *PEROXIDASE62* (*PRX62*, AT5G39580), where three additional SNPs located in

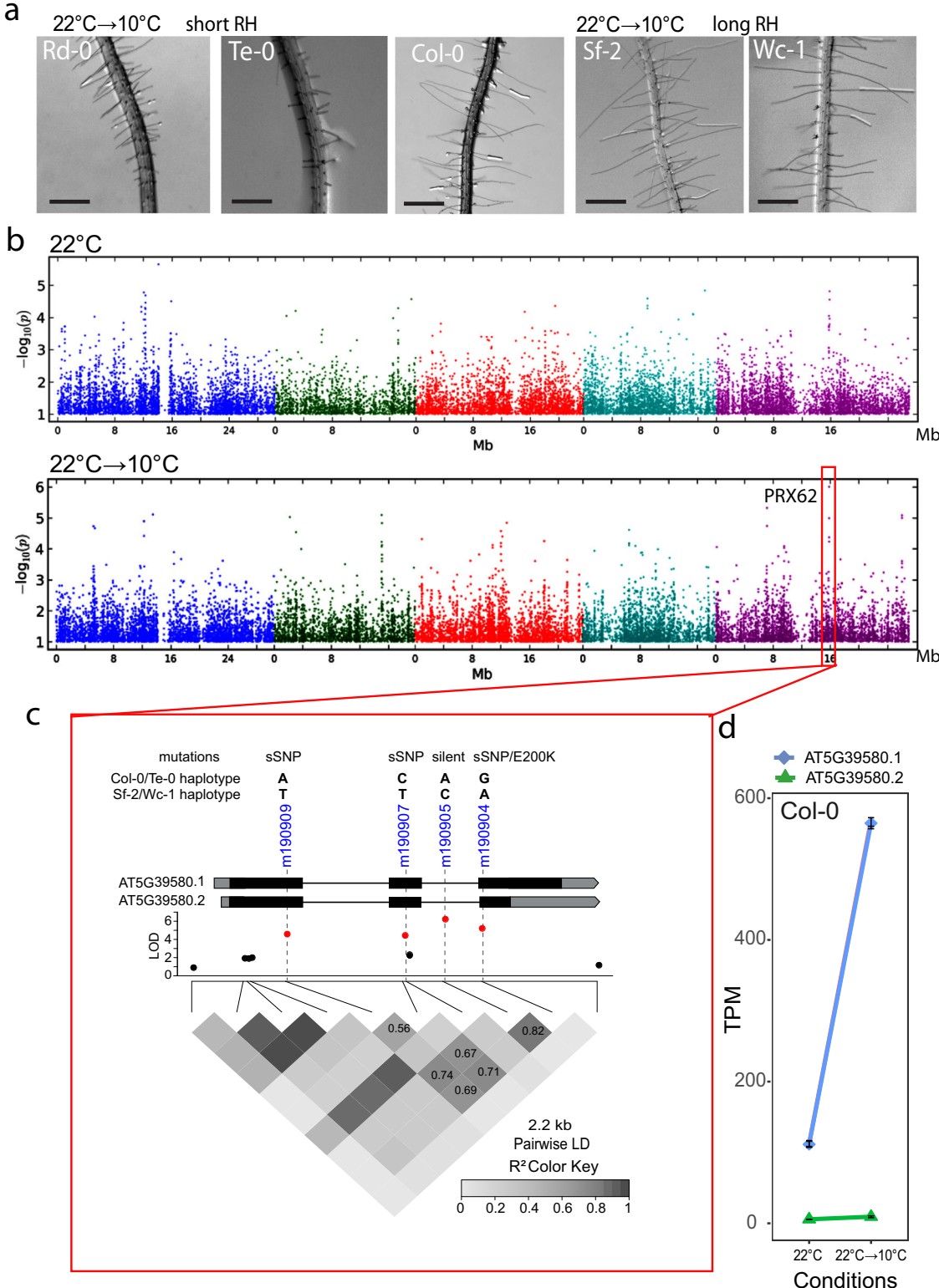

*PRX62* exons are in high linkage disequilibrium (LD) with m190905: m190904/15847644, m190907/15848071 and m190909/15848704 (LOD ~ 4.99–4.24, $r^2$ > 0.7, $p$ < 0.001, Fig. 1c). Interestingly, m190904 is encoding a non-synonymous SNP (nsSNP) only for the second splice variant of *PRX62* (AT5G39580.2). To test whether the association found is due to the amino acid change in AT5G39580.2, we first checked the expression pattern of both isoforms in *Arabidopsis thaliana* grown at 22 °C and at

22 °C→10 °C. Only the full-length transcript of *PRX62* (AT5G39580.1) is detectable in Col-0 and it increased up to 2.54 $\log_2$FC in roots exposed to low temperature (RNA-seq data, Fig. 1d). This was further confirmed by qPCR (Supplementary Fig. 3a). This result suggests that only AT5G39580.1 isoform is active and hence, the nsSNP on AT5G39580.2 is not the causal mutation for the association with *PRX62*. Therefore, we further analyzed the haplotypes formed by the four SNPs in LD (m190904,

**Fig. 1 PRX62 associates with enhanced RH growth under low temperature condition. a** Representative accessions of *A. thaliana* showing short (Rd-0 and Te-0) and long root hair (RH) phenotypes (Col-0, Sf-2, and Wc-1) when grown at low temperature (10 °C). Scale bars 500 μm **b** Manhattan plots for RH length at 22 °C (top plot) and at 10 °C (bottom plot). Coarse analysis was performed using GWAPP[29] (https://gwapp.gmi.oeaw.ac.at/). Arabidopsis chromosomes are depicted in different colors. The red box in the bottom plot indicates the genomic region significantly associated with root hair length at 10 °C. **c** Zoomed-in of the genomic region red-boxed in (**b**). The lead SNP (m190905) and three additional SNPs highly associated with RH length localize within *PRX62* (AT5G39580). *PRX62* splice variants (AT5G39580.1, AT5G39580.2) are depicted. The four associated SNPs (in red) are in high linkage disequilibrium (LD) with each other, and they are combined into two major and opposite haplotypes in the population (ACAG and TTCA). LD plot as heat-map and values for LD (coefficient of correlation, $r^2$) are shown. *PRX62* gene structure showing two splice variants (isoforms), common haplotypes, and type of mutation for AT5G39580.1/ AT5G39580.2 are indicated above the Manhattan plot. The SNP m190904 is a non-synonymous SNP for AT5G39580.2 causing a change from Glutamic Acid to Lysine at position 200 (E200K) in the amino acid sequence. This same SNP is synonymous (sSNP) for the full-length isoform AT5G39580.1. **d** The full-length variant of *PRX* (AT5G39580.1) is upregulated at low temperature (10 °C) while the shorter variant (AT5G39580.2) is almost not detected. Expression measured by RNA-seq of *PRX62*. TPM = Transcripts Per Kilobase Million.

m190905, m190907, and m190909). These SNPs combined into seven haplotypes in our population (*n* = 107), with two major allele-opposite haplotypes (ACAG, *n* = 80; TTCA, *n* = 16), two haplotypes with very low frequency (TCAG, *n* = 4; TCCA, *n* = 4) and three unique haplotypes (ACCG, TCCG, and ACAA). Analysis of variance between the average trait values for all non-unique haplotypes showed that RH length varies among them, being the first (ACAG) and second (TTCA) most frequent haplotypes significantly different for RH length (Supplementary Fig. 4). Since these are all synonymous SNPs in the active isoform AT5G39580.1, we speculate that the causal mutation was not scored in GWAS, but it is in LD with these four SNPs. Altogether, our results hinted at *PRX62* as a potential key factor in the regulation of RH growth under low temperature. The association found for *PRX62* with RH length at 10 °C explained 21% of the variance for this trait. Hence, as with many other complex traits, there are likely additional loci involved in RH length variation that failed to be detected by GWAS. Considering other candidate genes involved in RH-growth response at moderate low temperature datasets of whole seedlings[30] revealed that six PRX coding genes are induced at 10 °C; and notably, *PRX62* and *PEROXIDASE69* (*PRX69* AT5G64100) genes were predicted to be highly expressed in RHs (Supplementary Table 1). *PRX69* also has two variants, the full length AT5G64100.1 and a shorter one AT5G64100.2. By RNA-seq we also confirmed that only the full-length variant of *PRX69* is the most expressed one with a small upregulation (by 0.21 log$_2$FC) by low temperature although with similar overall transcriptional levels to *PRX62* (Fig. 2d). This was also confirmed by qPCR (Supplementary Fig. 3b). Therefore, we decided to characterize in depth both PRXs coding genes, *PRX62* and *PRX69* and their roles in RH growth at 10 °C.

In agreement with GWAS results, low temperature-mediated growth requires peroxidase activity since the treatment with salicylhydroxamic acid (SHAM), a peroxidase inhibitor[31,32] at inhibitory concentration 50% (IC$_{50}$ = 65 μM) at 22 °C, was able to repress up to 90% of this growth response at low temperature (Fig. 2a, b). Accordingly, peroxidase activity in whole roots was significantly lower under the SHAM treatments at both temperatures (Fig. 2c). We then tested if *PRX62* and *PRX69* expression levels were different between contrasting accessions based on the RH phenotype at 10 °C (Fig. 2d; Supplementary Fig. 3). Transcript levels of *PRX62* (after 3 days at 10 °C) coincided with the RH length of the given accession, i.e., the higher the expression of *PRX62* at 10 °C, the longer the RHs. This implies that high transcript levels of *PRX62* in Wc-1 and very low transcript levels in Bu-2 accessions might be linked to the differential RH phenotype detected at low temperature. Furthermore, this result reinforced our hypothesis that the causal variation for RH length differences is modulating *PRX62* expression (Fig. 2d; Supplementary Fig. 3). Since SNPs in the promoter region of *PRX62* were not scored in the initial GWAS, we mined 1001 *Arabidopsis* genomes (http://signal.salk.edu/

atg1001/3.0/gebrowser.php) and downloaded *PRX62* genomic sequences including ca. 2000 bp upstream the start site for 852 accessions (TAIR10 positions 15847080–15851079). After alignment (Supplementary File 3), we calculated the LD for the complete region. Our goal was to check whether the high-LOD SNPs in the coding region (m190904–m190909) are linked to SNPs in the promoter region. In fact, we identified four SNPs in the promoter of *PRX62* at −35, −101, −417, and −489 bp from the ATG, in significant LD with them ($r^2 > 0.1$, $p \lll 0.001$, Supplementary Fig. 5a, b). We deepened on the role of those SNPs for 60 accessions with combined genotypic and phenotypic information available (Supplementary Data 1). Significant differences in RH length were found for alleles at SNP-35 and for the allelic combination of SNP-35/m190905 (Supplementary Fig. 5c, d). In summary, GWAS originally detected a SNP in the intron of *PRX62*, but LD, sequencing and transcript analyses pointed to variation in the promoter region of *PRX62*, most likely including SNP-35 as one of the causal mutations. These findings explain the RH-length variation at 10 °C and the correlation between RH length and expression levels of *PRX62* in natural accessions. On the contrary, *PRX69* transcript levels are higher at 10 °C, but they did not show any significant variation across accessions, revealing an accession-independent activation of *PRX69* at low temperature. Altogether, these results suggest that upregulation of *PRX62* transcript levels together with high levels of *PRX69* may play an important role in RH growth at low temperature.

**PRX62 and PRX69 regulate RH growth under low temperature.** The in silico analysis of *PRX62* and *PRX69* expression using Tissue Specific Root eFP (http://bar.utoronto.ca/eplant/)[33] showed that both PRXs coding genes were confined to differentiated RH cells with expression in the elongation phase at similar levels than the RH marker *EXPANSIN 7* (Fig. 3a). Accordingly, the corresponding reporter lines of *PRX62$_{pro}$GFP*, as well as, *PRX69$_{pro}$GFP* showed high levels of signal in RH cells when grown at 10 °C while lower expression was detected at 22 °C (Fig. 3b). When *PRX62* and *PRX69* tagged constructs (*35S$_{pro}$PRX62-tagRFP* and *35S$_{pro}$PRX69-tagRFP*) are transiently coexpressed in *Nicotiana benthamiana* leaves with a plasma membrane marker, both PRXs showed an apoplastic localization (Supplementary Fig. 6). Overall, these results confirm that PRX62 and PRX69 are both cold-responsive specific RH class III PRXs that are secreted to the apoplastic space in the cell wall. At the coding sequence level, *PRX62* and *PRX69* belong to the same clade together with other three PRXs in a phylogenetic tree of the 75 apoplastic Class-III PRXs family members (Supplementary Fig. 7a). At protein sequence level both PRXs share more than 50% of sequence identity (Supplementary Fig. 7b). Outside *PRX62* and *PRX69*, the other three PRXs coding genes located in the same clade are not expressed in RH cells (Supplementary

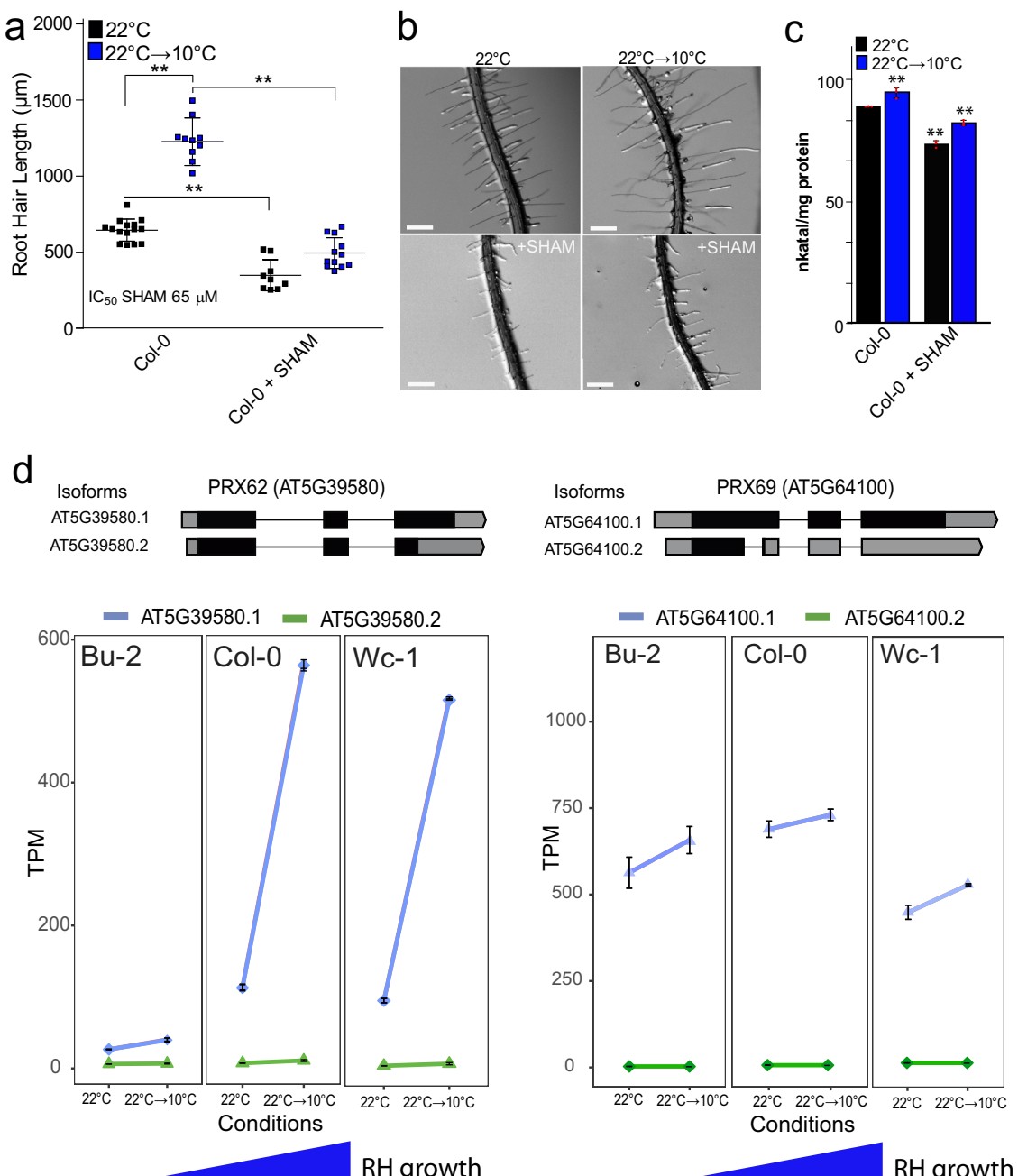

**Fig. 2 Low temperature enhanced RH growth requires peroxidase activity and upregulates *PEROXIDASE 62 (PRX62)* expression. a** RH length phenotype of Col-0 with or without the addition of the PRX inhibitor SHAM. Inhibitory Concentration 50 ($IC_{50}$) of RH grown at 22 °C was used (65 μM). Each point is the mean of the length of the 10 longest RHs identified in a single root. Data are the mean ± SD ($N = 16$ roots), two-way ANOVA followed by a Tukey–Kramer test; (**) $p < 0.01$. Results are representative of three independent experiments. Asterisks indicate significant differences between Col-0 and the corresponding genotype at the same temperature. Exact *p-values* are provided in the Source Data file. **b** Representative images of RH phenotype of Col-0 quantified in (**a**). Scale bars 500 μm. **c** Total root peroxidase activity. Peroxidase activity was assayed using guaiacol/hydrogen peroxide as substrate in root tissues grown with or without 65 μM SHAM, either for 5 days at 22 °C or for 5 days at 22 °C plus 3 days at 10 °C. Enzyme activity values (expressed as nkatal/mg protein) are the mean of three biological replicates ± SD. *p*-value of two-way ANNOVA followed by a Tukey–Kramer test (**) $p < 0.01$. Exact *p-values* are provided in the Source Data file. **d** In contrast to *PRX69*, *PRX62* is differentially expressed at low temperature (10 °C) in *Arabidopsis* accessions with contrasting RH phenotypes. Expression measured by RNA-seq of *PRX62* and *PRX69* in three contrasting *Arabidopsis* accessions based on the RH phenotype (short RH in Bu-2 and extra-long RH in Col-0 and Wc-1) detected at 10 °C. Isoforms' schemes were adapted from boxify (https://boxify.boku.ac.at/). TPM = Transcripts Per Kilobase Million.

Fig. 7a). In order to test if the absence of PRX62 and PRX69 is able to modify growth response at 10 °C, we assessed two T-DNA mutants for *PRX62* in the Col-0 background (*prx62-1* and *prx62-2*), being a knock-out (*prx62-1*) and a knock-down (*prx62-2*)

allele, respectively[34]. In addition, we also characterized two previously reported T-DNA mutants for *PRX69* (*prx69-1* and *prx69-2*)[34]. By RNA-seq, we confirm they were absence of transcripts for both *PRX62* and *PRX69* in these mutants (Supplementary

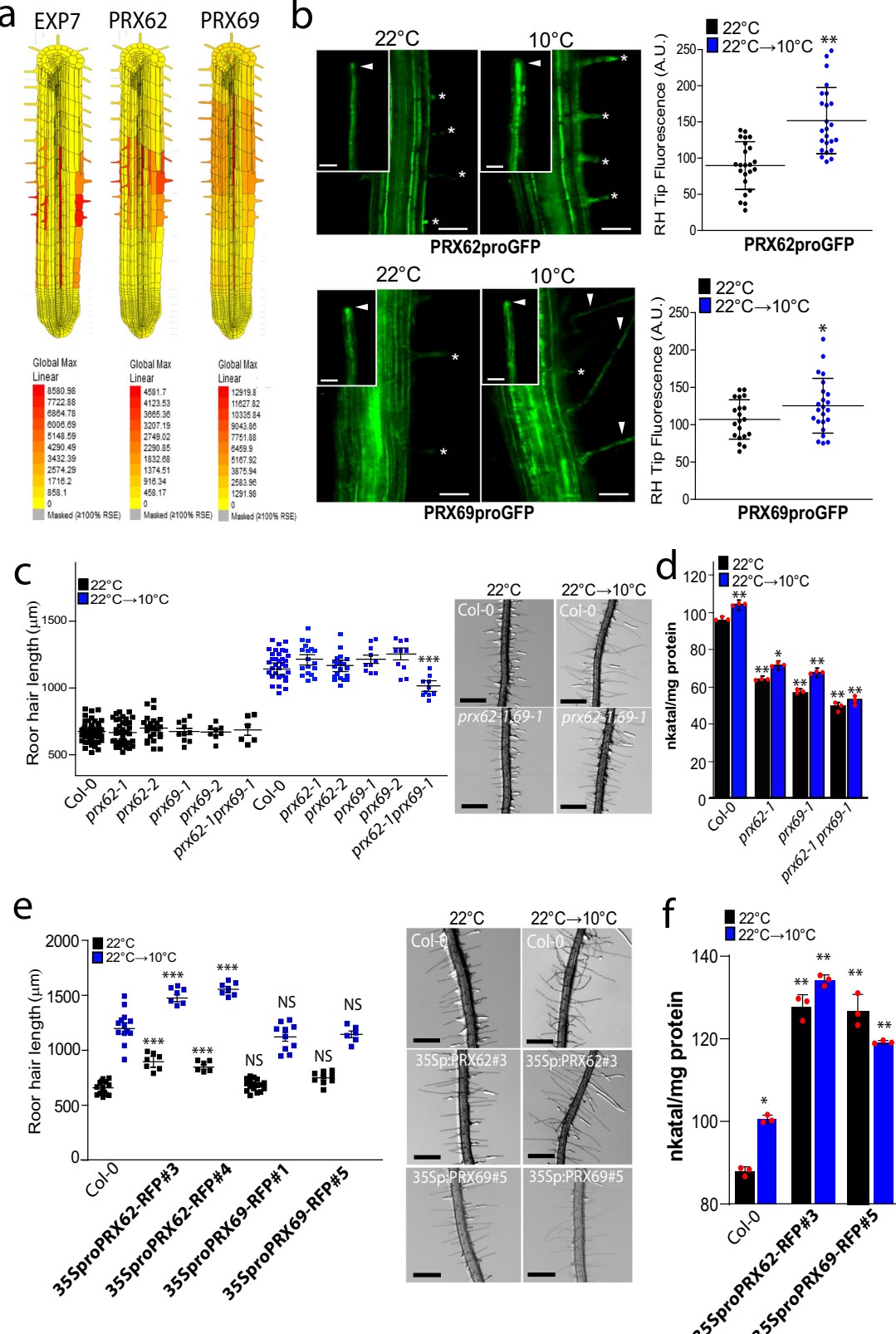

Fig. 8a). Only in *prx69-1* when grown at 10 °C we found a truncated transcript of *PRX69* (Supplementary Fig. 8b). The RH phenotype in both *prx62* and *prx69* single mutants were similar to Col-0 at 22 °C and at 10 °C (Fig. 3c) while the double mutant *prx62-1 prx69-1* showed significantly shorter RHs than Col-0 and

any of the PRXs single mutants at 10 °C. The double *prx62-1 prx69-1* mutant showed no detectable transcript levels of both PRXs genes (Supplementary Fig. 8c). The overall peroxidase activity was also partially impaired in single mutants *prx62-1* and *prx69-1* and double mutant *prx62-1 prx69-1* at both growth

**Fig. 3 *PEROXIDASE 62* (*PRX62*) and *PEROXIDASE 69* (*PRX69*) regulate RH growth and peroxidase activity under low temperature conditions. a** The *in silico* analysis of *PRX62* and *PRX69* genes expression using Tissue Specific Root eFP (http://bar.utoronto.ca/eplant/). The RH marker *EXPANSIN 7* was included for comparison. **b** Transcriptional reporters of *PRX62* (*PRX62_proGFP*) and *PRX69* (*PRX69_proGFP*) in the root elongation zone and specifically in RHs (RH) grown at 22 °C or 10 °C. Scale bar = 200 μm. Growing RHs are indicated with asterisks while already grown RHs with arrowheads. On the right, GFP signal is quantified. Each point is the signal derived from a single RH tip. Fluorescence AU data are the mean ± SD ($N = 20$ root hairs) according to Mann–Whitney U test; two-tailed $p$ value < 0.05. Results are representative of two independent experiments. Asterisks indicate significant differences between temperatures. Exact *p-values* are provided in the Source Data file. **c** Scatter-plot of RH length of Col-0, *PRX62* mutants (*prx62-1* and *prx62-2*) and *PRX69* mutants (*prx69-1* and *prx69-2*) and double mutant *prx62-1 prx69-1* grown at 22 °C or at 10 °C. Each point is the mean of the length of the 10 longest RHs identified in a single root. Data are the mean ± SD ($N = 10$ roots), two-way ANOVA followed by a Tukey–Kramer test; (***) $p < 0.001$. Results are representative of three independent experiments. Asterisks indicate significant differences between Col-0 and the corresponding genotype at the same temperature. Scale bars = 500 μm. Exact *p-values* are provided in the Source Data file. **d** Peroxidase activity was assayed using guaiacol/hydrogen peroxide as substrate in root tissues from Col-0, *prx62-1* and *prx69-1* seedlings grown either at 22 °C or at 10 °C. Enzyme activity values (expressed as nkatal/mg protein) are the mean of three biological replicates ± SD. $p$-value of two-way ANOVA followed by a Tukey–Kramer test, (**) $p < 0.01$, (*) $p < 0.05$. Exact *p-values* are provided in the Source Data file. **e** Scatter-plot of RH length of Col-0, *35S_proPRX62-tagRFP*/Col-0 and *35S_proPRX69-tagRFP*/Col-0 lines. Each point is the mean of the length of the 10 longest RHs identified in a single root. Results are the mean of three biological replicates ± SD ($N = 10$ roots). Asterisks indicate significant differences between Col-0 and the corresponding genotype at the same temperature (two-way ANOVA followed by a Tukey–Kramer test; (***) $p < 0.001$). NS = non-significant differences. Exact *p-values* are provided in the Source Data file. Scale bars = 500 μm. **f** Peroxidase activity was assayed using guaiacol/hydrogen peroxide as substrate in root tissues from Col-0 and overexpression lines *35S_proPRX62-tagRFP*/Col-0 #3 and *35S_proPRX69-tagRFP*/Col-0 #5 grown either at 22 °C or at 10 °C. Enzyme activity values (expressed as nkatal/mg protein) are the mean of three biological replicates ± SD. $p$-value of two-way ANOVA followed by a Tukey–Kramer test, (**) $p < 0.01$, (*) $p < 0.05$. Exact *p-values* are provided in the Source Data file.

temperatures, 22 and 10 °C (Fig. 3d). We then tested RH growth complementation of the *prx62-1 prx69-1* double mutant by expressing either *PRX62* or *PRX69* coding sequences under 35S promoter (*35S_proPRX62*, *35S_proPRX69*). The RH growth was restored comparable to Col-0 levels at 10 °C only for *PRX62* but not for *PRX69* (Supplementary Fig. 9a, b). This suggests that high levels of *PRX62* but not of *PRX69* are able to compensate the absence of both PRXs in *prx62-1 prx69-1* double mutant. To determine whether higher expression of the PRX62 and PRX69-encoding genes are sufficient to trigger changes in RH cell length, we generated a constitutive *35S_proPRX62* overexpression lines in the Col-0 background that expressed up to 13–52 folds of transcripts levels of *PRX62* as well as the corresponding *35S_proPRX69* overexpression lines with 9–11 folds (Supplementary Fig. 8c). As expected, *PRX62* overexpression resulted in significantly longer RH cells than their respective Col-0 while *PRX69* overexpression failed to trigger enhanced growth (Fig. 3e). This may indicate that PRX62 and PRX69 do not have equal functions in RH growth although both PRXs are required for this enhanced low temperature growth process. The overall peroxidase activity in both, *35S_proPRX62* and *35S_proPRX69* was significantly higher than in Wt Col-0 at both growth temperatures, 22 and 10 °C (Fig. 3f). Taken together, these results indicate that the amount of PRX62 protein linked to its peroxidase activity control RH growth at 10 °C, in contrast to PRX69. Finally, we tested if SHAM may further inhibit the RH growth in the *prx62-1 prx69-1* double mutant at 10 °C (Supplementary Fig. 10). As expected, a strong inhibition was detected not only in Wt Col-0 but also in the *prx62-1 prx69-1* double mutant suggesting that other PRXs might be also active during RH growth at 10 °C. This underscores a high degree of genetic redundancy on PRXs acting on RH growth as previously detected for roots growing on low nutrients at room temperature[2,13].

**The absence of PRX62 and PRX69 proteins induced a deregulation of several PRXs and cell wall EXTs genes at low temperature.** To better understand the transcriptional changes produced at low temperature in a *PRX62*- and *PRX69*-dependent manner, we performed an RNA-seq analysis comparing Col-0 and the double *prx62-1 prx69-1* mutant at 10 °C or 22 °C. We found a central core of 1551 differentially expressed genes (DEG) at low temperature grouped into 10 clusters that were

misregulated in the double *prx62-1 prx69-1* mutant compared to Col-0 (Fig. 4a). 1053 genes were upregulated (clusters 1–6) and 498 were downregulated (clusters 7–10) in Col-0 compared to the double *prx62-1 prx69* mutant-*1*. We focused on the largest clusters 1, 2, and cluster 4 (comprised by 875 genes) where the genes upregulated in Col-0 were deregulated in double *prx62-1 prx69-1* mutant in response to cold. In these gene clusters, over-represented GO terms were linked to plant cell walls, extracellular domains, and secretory pathway (Fig. 4a). We identified several over-represented *PRXs* (15 genes) and EXTs-related proteins (7 encoding genes) suggesting a global change in ROS-homeostasis and EXTs cell wall remodeling in the double *prx62-1 prx69-1* mutant at low temperature (Fig. 4a). Some of these genes (e.g., *EXT6* and *PRP1*) showed a gene dose-dependent expression at transcript level linked to the RH growth phenotype at 10 °C (Fig. 4b). This indicated that low temperature induces global gene expression changes linked to the cell wall remodeling and ROS-homeostasis that positively enhance RH growth. The analysis highlights that the absence of PRX62 and PRX69 proteins triggers major changes in the transcriptional program of other PRXs and EXTs genes at low temperature. This implies the existence of a feedback regulatory loop from the apoplast-cell wall compartments that triggers major changes at the transcriptional level of cell wall proteins and apoplastic PRXs.

**PRX62 and PRX69 affect ROS-homeostasis in RH cells under low temperature.** To get a deeper insight into PRX62 and PRX69 protein functions in growing RHs at moderate low temperature, we explored the effect of these PRXs on Reactive Oxygen Species (ROS)-homeostasis. Overall PRX functions are linked to ROS, which are one of the key factors regulating polar growth in RHs[2,4,35]. Then, we measured total cytoplasmic ROS ($_{cyt}$ROS) using the cell-permeable fluorogenic probe 2′,7′-dichlorodihydro-drofluorescein diacetate (H$_2$DCF-DA) and apoplastic ROS ($_{apo}$ROS) levels with cell-impermeable Amplex™ UltraRed Reagent in RH tips at 22 and 10 °C (Fig. 5a, b). The double mutant *prx62-1 prx69-1* showed higher levels of $_{cyt}$ROS at 10 °C in actively growing RH tips compared to Col-0 (Fig. 5a) while this enhancement in ROS level is less evident at 22 °C between the double mutant *prx62-1 prx69-1* and Col-0. In agreement, in the plants overexpressing PRX62 or PRX69, $_{cyt}$ROS were reduced at both 22 and 10 °C. On the other hand, the $_{apo}$ROS in the RH tip

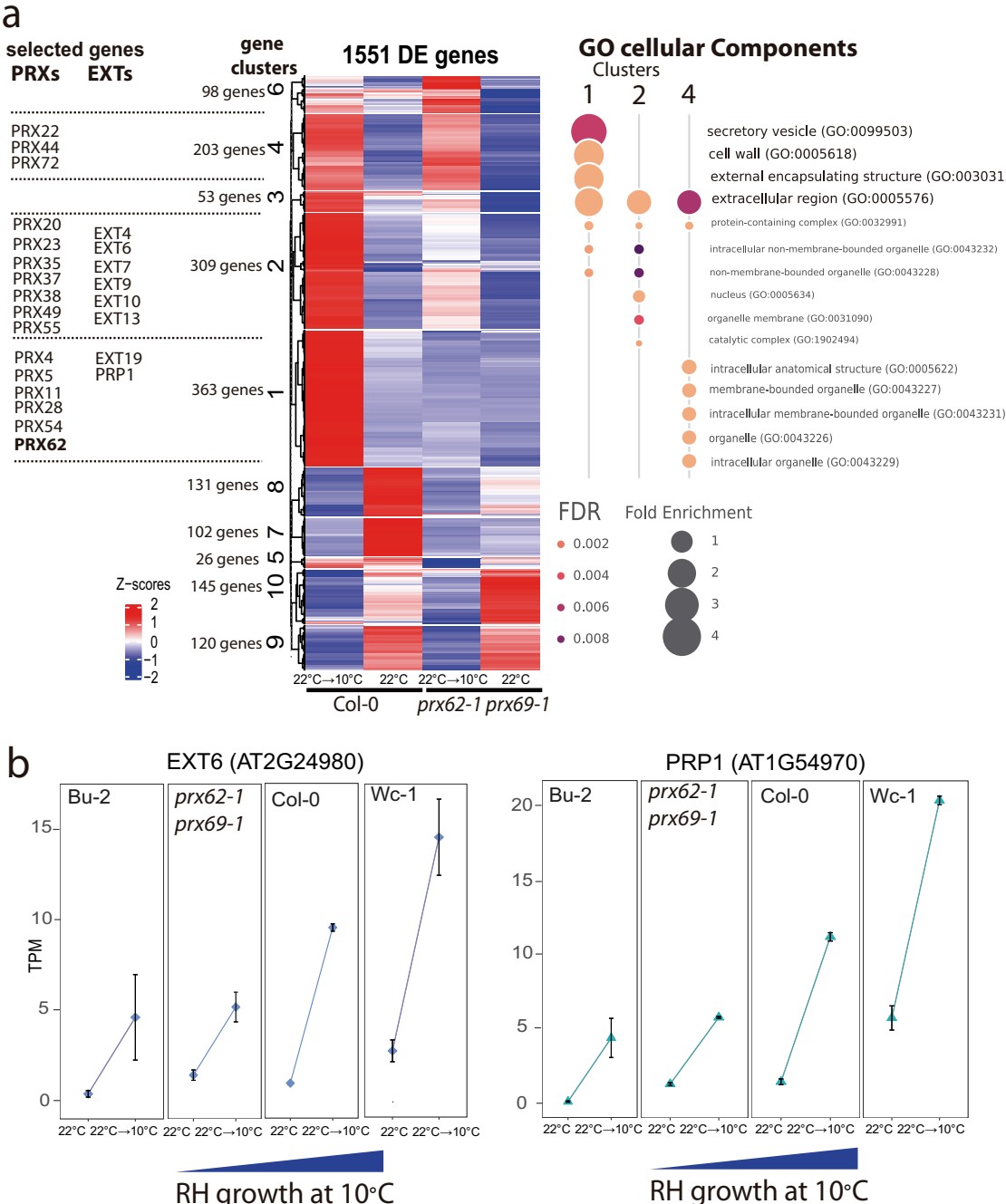

**Fig. 4 Global transcriptomic changes induced by low temperature and misregulated in the *prx62-1 prx69-1* mutant are associated with PRXs and cell associated-EXTs. a** Heat-map showing the hierarchical gene clustering for 1551 *A. thaliana* genes differentially expressed (DE) between room temperature growth (22 °C) and low temperature (10 °C) growth in wild type Col-0 and in double mutant *prx62-1 prx69-1* roots. Gene Ontology analysis results depicting the top 7 most significantly enriched GO terms are shown as bubble plots on the right for the clusters of interest. DE genes in clusters 1, 2 and 4 were contrasted against all the expressed genes for GO analysis. The size of the points reflects the amount of gene numbers enriched in the GO term. The color of the points means the *p*-value. Relevant gene examples of specific clusters (1, 2, and 4) are listed on the left. **b** Expression of *EXT6* and *PRP1* is gradually upregulated at low temperature (10 °C) in 4 genotypes from very short RHs (Bu-2) to very long RHs (Wc-1) (RNA-seq data). TPM = Transcripts Per Kilobase Million.

were enhanced in Col-0 at 10 °C compared to the levels at 22 °C while they were lower in the double mutant *prx62-1 prx69-1* at both temperatures. In the lines overexpressing PRX62 or PRX69 coding genes, $_{apo}$ROS were enhanced at both 22 and 10 °C (Fig. 5b). The increased level of $_{apo}$ROS in Col-0 under low temperature is in agreement with a two-fold increase in the transcript levels for *NOXC* (*RBOHC*), a key enzyme-encoding gene for ROS production[36,37] (Supplementary Fig. 11).

Collectively, these results suggest that ROS-homeostasis is drastically modified in an antagonistic manner by the absence or overexpression of these two PRXs when RH grow at 10 °C, affecting RH cell elongation.

**Low temperature enhances EXTENSIN cell wall insolubility in RH cells.** EXT-crosslinking can provide architectural stabilization for

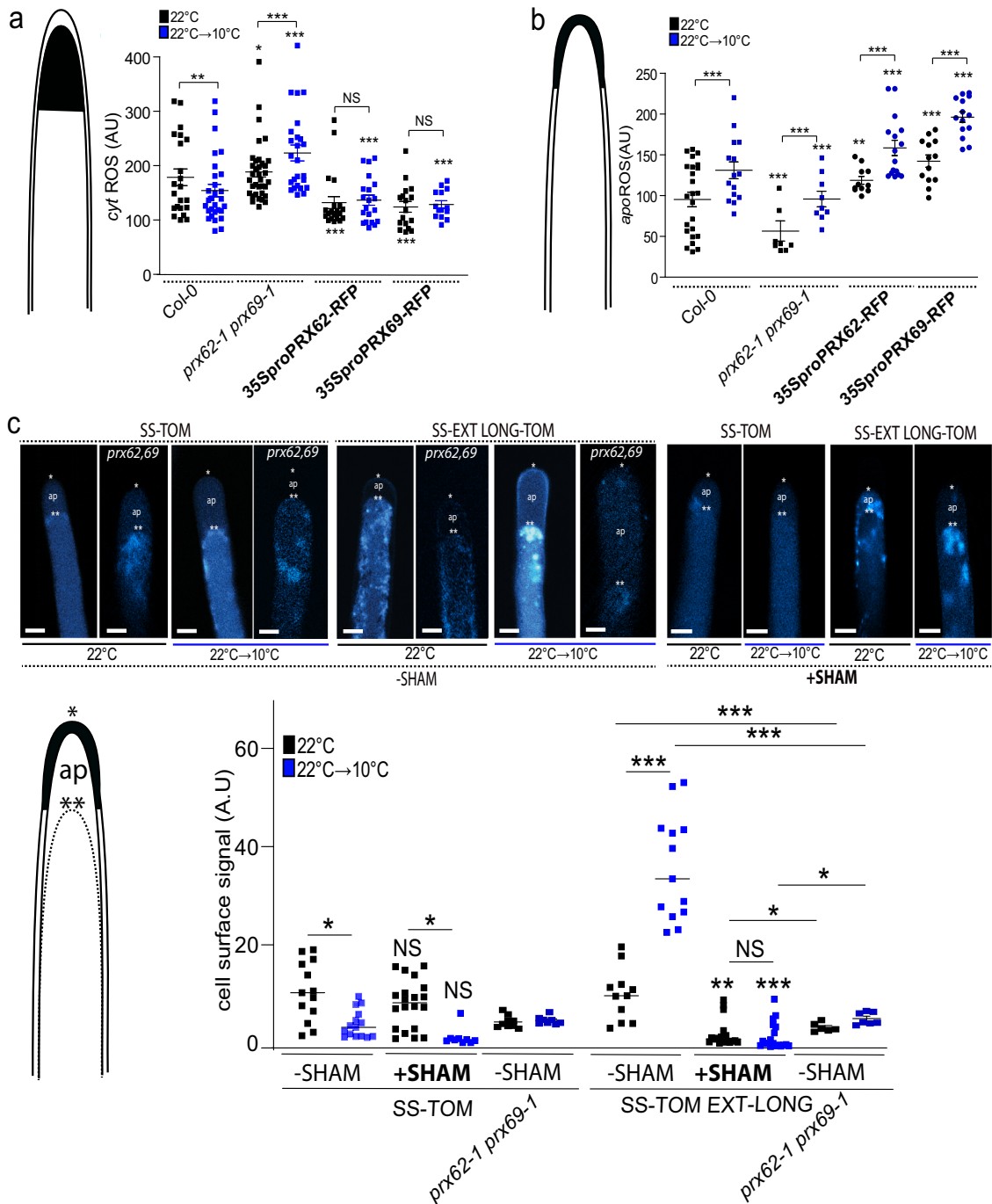

normal wall reinforcement during cell elongation[38–42]. Since changes in ROS-homeostasis could lead to abnormal cell wall secretion and structure, we wondered whether PRX62 and PRX69 might participate in the cell wall glycoprotein EXTs crosslinking during RH growth at low temperature. Then, we tested if low temperature could induce a change in the targeting of EXTs secreted and insolubilized in the wall by the activity of these two PRXs. To this end, we used an EXT-reporter carrying a tdTomato tag (SS-TOM-Long-EXT) that is resistant to acidic pH, a condition usually found in the cell wall-apoplast compartments, and a secreted tdTomato tag (SS-TOM) was used as a control (Supplementary Fig. 12)[13]. The signal coming from the cell surface in the apical zones of RHs cells under plasmolysis conditions were determined for SS-TOM-Long-EXT and SS-TOM constructs at 22 °C/10 °C temperatures and SHAM-treated/non-treated roots (Fig. 5c). Plasmolysis allowed us to retract the plasma membrane and define the EXT-signal coming specifically from the cell walls. Interestingly, of cell wall stabilization/insolubility of SS-TOM-Long-EXT in the RH tip was drastically augmented at 10 °C compared to 22 °C. Furthermore, the signal increment at 10 °C was completely abolished when roots were treated with the peroxidase inhibitor SHAM (Fig. 5c). Thus, the SS-TOM-Long-EXT reporter tested in the apical zone of the RHs is modified by low temperature and by the peroxidase activity, at least partially possibly exerted by PRX62 and PRX69 in the apoplast. To assess this hypothesis, we expressed both reporters, SS-TOM-Long-EXT and SS-TOM, in the prx62-1 prx69-1 double mutant, and a similar analysis was carried out (Fig. 5c). Significantly lower signal was detected as insolubilized SS-TOM-EXT-Long reporter in the prx62-1 prx69-1 double mutant when compared to Wt Col-0 while similar signal was quantified for the control SS-TOM in both genotypes.

**Fig. 5 ROS-homeostasis and EXT-stabilization in RH apical cell wall depend on PRX62 and PRX69 proteins. a** Cytoplasmic ROS ($_{cyt}$ROS) levels were measured using 2′,7′-dichlorodihydrofluorescein diacetate (H$_2$DCF-DA) in apical areas of RHs in wild-type (Columbia Col-0), in the double mutant *prx62-1 prx69-1* and in the *35S$_{pro}$PRX62-tagRFP*/Col-0 and *35S$_{pro}$PRX69-tagRFP*/Col-0 lines grown at 22 °C or 10 °C. Each point is the signal derived from a single RH tip. Fluorescence AU data are the mean ± SD ($N = 18$ root hairs), two-way ANOVA followed by a Tukey–Kramer test; (**) $p < 0.01$ (***) $p < 0.001$. Results are representative of two independent experiments. Asterisks indicate significant differences. NS = non-significant differences. Exact *p-values* are provided in the Source Data file. **b** Apoplastic ROS ($_{apo}$ROS) levels were measured with Amplex™ UltraRed in apical areas of RHs in wild-type (Columbia Col-0), in the double mutant *prx62-1 prx69-1* and in the *35S$_{pro}$PRX62-tagRFP*/Col-0 and *35S$_{pro}$PRX69-tagRFP*/Col-0 lines grown at 22 °C or 10 °C. Each point is the signal derived from a single RH tip. Fluorescence AU data are the mean ± SD ($N = 11$ root hairs), two-way ANOVA followed by a Tukey–Kramer test; (***) $p < 0.001$. Results are representative of two independent experiments. Asterisks indicate significant differences. Exact *p*-values are provided in the Source Data file. **c** Signal of SS-TOM and SS-EXT LONG-TOM in the apical zone of RHs grown at 22 °C or 10 °C with or without SHAM treatment in Col-0 and in the *prx62-1 prx69-1* double mutant without SHAM. Cells were plasmolyzed with a mannitol 8% solution. In the images: (*) indicates cell surface including the plant cell walls, (**) indicates the retraction of the plasma membrane, (ap) apoplastic space delimitated between the plant cell wall and the retracted plasma membrane. Each point is the signal derived from a single RH tip. Fluorescence AU data are the mean ± SD SD ($N = 11$ root hairs), two-way ANOVA followed by a Tukey–Kramer test; (*) $p < 0.05$, (**) $p < 0.01$, (***) $p < 0.001$. Results are representative of two independent experiments. Asterisks on the graph indicate significant differences between each genotype 22 °C vs 10 °C. NS = non-significant differences. Exact *p*-values are provided in the Source Data file. Scale bars = 5 µm.

Finally, to postulate that these two PRXs (PRX62 and PRX69) might be able to interact with single-chain EXTs, we performed homology modeling with GvEP1, an EXT-PRX that is able to crosslink EXTs in vitro[43,44]. By *docking* analysis, we obtained interaction energies (Kcal/mol) with four different short EXT peptides: a non-hydroxylated peptide, a hydroxylated peptide, an arabinosylated peptide, and an arabino-galactosylated peptide (Supplementary Fig. 13). It was previously shown that mutants carrying under-*O*-glycosylated EXTs have severe defects in root hair growth[6,45]. Our docking results for the two PRXs show small but consistent interaction energy differences that depend on the EXT glycosylation state, being higher for non-*O*-glycosylated species. In addition, *O*-glycosylated EXT variants docked in a rather dispersed way while non-*O*-glycosylated variants preferentially docked in a grooved area (Supplementary Fig. 13). Furthermore, EXT substrate binds in a groove very close to the heme, which is partially accessible to solvent and, thus, potentially able to react with the substrate. Overall, our analysis indicates that PRX62 and PRX69 might interact with EXT substrates in open regions of the EXT backbones with little or no *O*-glycosylation. This is in agreement with previous studies that suggested that high levels of *O*-glycosylation in certain EXT segments physically restrict EXT lateral alignments, possibly by acting as a branching point[6,35,38]. All together, these results (Fig. 5 and Supplementary Fig. 13) suggests that changes in ROS-homeostasis produced by altered levels of these PRXs in the apoplast might affect the secretion, targeting and, possibly the crosslinking of cell wall components including EXTs, affecting RH cell elongation (Supplementary Fig. 14).

## Discussion

Despite the putative high overall genetic redundant functions of apoplastic Class-III PRXs, in the last years several individual PRXs were characterized to be involved in the oxidative polymerization of monolignols in the apoplast of the lignifying cells in xylem (e.g., PRX17[46]), in the root endodermis (e.g., PRX64[47]) or in petal detachment[48]. Moreover, PRXs are also able to polymerize other components of the plant cell wall, including suberin, pectins, and EXTs[44,49–51]. While several candidates of PRXs have been described in diverse plants to be associated specifically to EXTs crosslinking (EXT-PRXs) by in vitro studies (LEP1, GvEP1, and FBP1) or immunolocalization evidences linked them to transient activity measurements (PRX08 and PRX34)[15,43,44,49,52–54], their role in vivo remains largely unexplored. Previously it was demonstrated that three PRXs (PRX01, PRX44, and PRX73) directly contribute to ROS-homeostasis and RH growth at room

temperature (22 °C) under low-nutrient condition[2,13]. By using a GWAS/RNA-seq approach, we identified here two previously poorly characterized apoplastic peroxidases, PRX62 and PRX69[34], as positive regulators of RH growth at low temperature (10 °C). One of the key results of this work is that *PRX62* gene was found using GWAS while *PRX69* was identified based on the transcriptomic profile. This may suggest a different evolutionary history for both proteins. PRX62 may have evolved to give a dose-response according to the allele encoded in the genome, while PRX69 has a constitutive response at low temperature. These features of PRX62 and PRX69 could be useful in crop improvement, to select varieties with differential responses; better adapted to the environment they are exposed. The evidences shown here indicate that PRX62 and PRX69, to a lesser extent, are involved in the ROS-homeostasis linked to the association of EXTs to the cell wall during RH cell elongation at low temperature (Supplementary Fig. 14). We speculate that cell wall insolubilization/association of EXTs triggered by low temperature might not only involve Tyr-covalent crosslinks mediated by these two PRXs identified here but also by EXT hydrophobic associations non-dependent on Tyr as suggested before for Leucine-Rich Extensins 1 (LRX1[55]). Further analyses might shed light on these complex processes.

Previously, our group as well as others have documented that changes in any of the several posttranslational modifications in EXTs and related-EXTs like LRXs (e.g., proline hydroxylation, *O*-glycosylation, and Tyr-crosslinking), all affected RH growth[6,13,35,45,56,57], as well as pollen tube growth[58–60]. In addition, auxin-dependent ROS-homeostasis controlled by three apoplastic PRXs (e.g., PRX01, PRX44, and PRX73) and plasma membrane RBOHC protein (also known as RHD2, for RH DEFECTIVE 2) was shown to be determinant for a proper RH growth under low nutrient condition[2,13] or under low temperature[28]. Collectively, these evidences highlight the predominant role of ROS-homeostasis partially regulated by specific PRXs as a key component in polar RH elongation. The molecular mechanism by which low temperature-associated nutrient availability in the media[27,28] triggers the expression of these two specific PRXs remains unclear, although RSL4 could play a central role in the regulation of this mechanism. Previously, we have shown that the lncRNA *APOLO* binds to the locus of RHD6 and controls *RHD6* transcriptional activity leading to cold-enhanced RH elongation through the consequent activation of *RSL4*[27] and of several cell wall EXTENSIN (EXT) encoding genes[28]. Unexpectedly, our previous results indicate that the low temperatures (10 °C) are able to trigger an exacerbated RH growth compared with cell expansion at room temperature[28]. Moreover, further

research will be needed to uncover the nutritional signal perceived at the RH cell surface to trigger PRX62 and PRX69 low temperature mediated growth response. The expression levels of *PRX62* and *PRX69* orthologs in other Brassicaceae could be used as biomarkers for crop improvement in the selection of genotypes with longer RHs at moderate low temperatures in order to boost nutrients uptake in deficient soils.

## Methods

**Plant genotyping and growth conditions.** *Arabidopsis thaliana* Columbia-0 (Col-0) was used as the wild type (Wt) genotype in all experiments unless stated otherwise. Seedlings were surface sterilized and stratified in darkness at 4 °C for 3 days before been germinated on ½ strength MS agar plates supplemented with MES (Duchefa, Netherlands), in a plant growth chamber in continuous light (120 μmol s⁻¹ m⁻²). Plants were transferred to soil for growth under the same conditions as previously described at 22 °C. Mutants and transgenic lines developed and used in this study are listed in Supplementary Table 2. For the identification of T-DNA knockout lines, genomic DNA was extracted from rosette leaves. Confirmation by PCR of a single and multiple T-DNA insertions in the genes were performed using an insertion-specific LBb1 or LBb1.3 (for SAIL or SALK lines, respectively) or 8474 (for GABI line) primer in addition to one gene-specific primer. In this way, we isolated homozygous for all the genes. Arabidopsis T-DNA insertions lines (*prx62-1* [GK_287E07], *prx62-2* [SALK_151762], *prx69-1* [SAIL_691_G12], *prx69-2* [SALK_137991]) were obtained from the European Arabidopsis Stock Center (http://arabidopsis.info/). Using standard procedures homozygous mutant plants were identified by PCR genotyping with the gene-specific primers listed in Supplementary Table 3. T-DNA insertion sites were confirmed by sequencing using the same primers. Plants were routinely grown in Jiffy peat pellets (continuous light, 120 μmol photons/m/s, 22 °C, 67% relative humidity). For in vitro experiments, seeds were surface-sterilized and sown in Petri dishes on agar-solidified half-MS medium without sucrose, and grown in a culture room with continuous light (120 μmol photons/m/s, 22 °C).

**Root hair phenotype.** Seeds were surface sterilized and stratified in darkness for 3 days at 4 °C. Then grown on ½ strength MS agar plates supplemented with MES (Duchefa, Netherlands), in a plant growth chamber at 22 °C in continuous light (120 μmol s⁻¹ m⁻²) for 5 days at 22 °C plus 3 days at 10 °C (moderate-low temperature treatment) or for 8 days at 22 °C as control. For quantitative analyses of RH phenotypes, 10 fully elongated RH from the maturation zone were measured per root under the same conditions from treatment and control. Measurements were made after 8 days. Images were captured using an Olympus SZX7 Zoom Stereo Microscope (Olympus, Japan) equipped with a Q-Colors digital camera and Q Capture Pro 7 software (Olympus, Japan). Results were expressed as the mean ± SD using the GraphPad Prism 8.0.1 (USA) statistical analysis software. Results are representative of three independent experiments, each involving 10–20 roots.

**GWAS, haplotype, and re-sequenced data analyses.** To perform GWAS, 108 *Arabidopsis thaliana* natural accessions were phenotyped for RH length in a shift-temperature experiment as described above (see Source Data file corresponding to Supplementary Fig. 2 for a full list of accessions). All accessions are publicly available and can be requested to the Arabidopsis Biological Resource Center (ABRC: https://abrc.osu.edu/) or to the Nottingham Arabidopsis Stock Center (NASC: http://arabidopsis.info/BasicForm) by common name or genotype ID. The population (except Bu-2 accession) was previously genotyped using 214,051 Single Nucleotide Polymorphisms (SNPs) and this information is publicly available[61]. The 107 pairs of phenotype/genotype were used to perform GWAS on the GWAPP web application from the GWA-Portal[29] (https://gwas.gmi.oeaw.ac.at/#/home, Experiment code: 3b316208-0b5d-11e7-b6b1-005056990049) applying the accelerated mixed model, AMM[62–64]. A total of 139,425 SNPs with minor allele frequency (MAF) ≥ 10% were retained for further analysis. *P*-values of association were log-transformed to LOD values (−log₁₀ (*p*-value)) and corrected for multiple comparisons using FDR procedure[65]. The threshold for significant associations was set to *p*-value ≤ 1/*N* (where *N* is the number of SNPs = 139,425). Manhattan plots were obtained using the qqman package[66] in R, filtering out the SNPs with *p*-value > 0.4, to minimize overrepresentation of non-significant SNPs. Linkage disequilibrium, i.e., the degree to which an allele of one SNP co-occurs with an allele of another SNP within a population, was calculated as the square coefficient of correlation (*r*²) and visualized using the LDheatmap package[67] in R. Haplotypes were analyzed using ANOVA followed by Tukey test implemented in Infostat[68]. Re-sequenced data from 852 accession was download, aligned, and used to calculate LD between SNPs in *PRX62* coding region and promoter region (http://signal.salk.edu/atg1001/3.0/gebrowser.php, TAIR10 positions 15847080-15851079). Alignment of the forward DNA strand (Supplementary Data 1) was performed using MAFFT online tool[69]. SNPs in the text are called as in the reverse complement strand, where *PRX62* is encoded, except for in the section of the Source Data file corresponding to Supplementary Fig. 2 that are called as in the forward DNA strand, to easily relate to SNPchip data.

**Peroxidase activity.** Soluble proteins were extracted from roots grown on vertical plates for 10 days at 22 °C or 10 °C by grinding in 20 mM HEPES, pH 7.0, containing 1 mM EGTA, 10 mM ascorbic acid, and PVP PolyclarAT (100 mg/g fresh material)(Sigma, Buchs, Switzerland). The extract was centrifuged twice for 10 min at 10,000×*g*. Each extract was assayed for protein levels with the Bio-Rad protein assay (Bio-Rad, USA). Enzyme activity (expressed in nkatal/mg protein) was determined at 25 °C by following the oxidation of 8 mM guaiacol (Fluka™, Honeywell International,USA) at 470 nm in the presence of 2 mM H₂O₂ (Carlo Erba, Italy) in a phosphate buffer (200 mM, pH6.0). Results were expressed as the mean ± SD. Results are representative of three independent experiments.

*Gene transcript analysis by quantitative PCR (qPCR).* Total RNA was prepared from 10 days old in vitro-grown plantlets using the TRI™ Reagent Solution (Sigma-Aldrich). After quantification by spectrophotometry and verification by electrophoresis, RNA was treated with the RQ1 RNase-free DNase I (Promega). One microgram of total RNA was reverse transcribed using an oligo(dT)₁₅ and the MMLV-RT (Promega) according to the manufacturer's instructions. cDNA was diluted 20-fold before PCR. qPCR was performed on a QuantStudio 6 Flex Real-Time PCR System (Thermo Fisher) using 5 μL Power SYBR Green PCR Mix (Applied Biosystems), 2 μL of cDNA, and 0.3 μM of each primer in a total volume of 10 μL per reaction. Primers used are listed in Supplementary Table 3. *ACT2* (AT3G18780) and *UBQ1* (AT3G52590) genes were used as references for normalization of gene expression levels. The cycling conditions were 95 °C for 10 min, 40 cycles of 95 °C for 15 s, 60 °C for 1 min. and finally a melting curve from 60 to 95 °C (0.05°/s). Under these conditions, primers efficiency was found to be between 97.0 and 99.7%. No amplification occurred in the no-template controls. Data were analyzed using the Standard curve method[70] and Qiagen REST© 2009 software[71]. Three independent experiments (and two or three biological and technical replicates per experiment) were performed.

**PRXs-tagged reporter lines.** For the *PRX62ₚᵣₒGFP* and *PRX69ₚᵣₒGFP* reporter lines, a 1.5 kb genomic region upstream of the ATG start codon of each *PRX62* (AT5G39580) and *PRX69* (AT5G64100) genes was selected using ThaleMine (https://bar.utoronto.ca/thalemine/begin.do) synthetized and cloned into the pUC57 vector by GenScript Biotech(USA). Through Gateway cloning Technology (Invitrogen) the 1.5 kb upstream regions were recombined first in pDONR™207 vector and subcloned into pMDC111 destination vector[72] (Invitrogen) for *PRX69* promoter region and into pGWB4 vector[73] (Invitrogen) for *PRX62* promoter region. These constructs were checked by restriction analysis. Both vectors contain a cassette with a C-terminal GFP tag. For the PRXs-tagRFP lines, the *PRX62* and *PRX69* coding sequence was amplified by PCR from *A. thaliana* 10-day old plantlets cDNAs using specific primers (Supplementary Table 3). The PCR product was digested with *Hind*III and *Bam*HI (*PRX62*) or with *Eco*RI and *Sma*I (*PRX69*), and cloned into Gateway® TagRFP-AS-N entry clone (Evrogen). The *PRX62-tagRFP* fusion was subcloned (Gateway Technology, Invitrogen) into the pB7WG2 destination vector[72] containing a 35S promoter. This construct was checked by restriction analysis and sequencing. The same procedure was used to generate *35SₚᵣₒPRX69-tagRFP* construct. All the constructs were used to transform *A. thaliana* plants and obtain homozygous stable fluorescent lines.

**Confocal microscopy.** Confocal laser scanning microscopy for the lines *PRX62ₚᵣₒGFP* and *PRX69ₚᵣₒGFP*, was performed using Zeiss LSM5 Pascal (Zeiss, Germany) (Excitation: 488 nm argon laser; Emission: 490–525 nm, Zeiss Plain Apochromat 10X/0.30 or 40X/1.2 WI objectives according to experiment purpose). Z stacks were done with an optical slice of 1 μm, and fluorescence intensity was measured at the RH tip. For the lines *SS-TOMATO*; *SS-TOMATO-EXT LONG*; *SS-TOMATO/prx62-1 prx69-1* and *SS-TOMATO-EXT LONG/prx62-1 prx69-1*, roots were plasmolyzed with a 8% mannitol solution, and the scanning was performed using Zeiss LSM 510 META (Zeiss, Germany)(Excitation: 543 nm argon laser; Emission: 560–600 nm, Zeiss Plain Apochromat 63X/1.4 -Oil objective). GFP signal and tdTOMATO cell wall signal at RH tip were quantified using the ImageJ software. Fluorescence AU was expressed as the mean ± SD using the GraphPad Prism 8.0.1 (USA) statistical analysis software. Results are representative of two independent experiments, each involving 10–15 roots, and approximately, between 10 and 20 hairs per root were observed.

**Apoplastic and cytoplasmic ROS measurement in RH Tip.** To measure ROS levels in root hairs cells, 8 days-old Arabidopsis seedlings grown at 22 °C (control) and 10 °C in continuous light were used. For cytoplasmic ROS, the seedlings were incubated in darkness for 10 min with 50 μM H2DCF-DA (Thermo Fisher) at room temperature then washed with liquid 0.5X MS media (Duchefa, Netherlands) and observed with Zeiss Imager A2 Epifluorescence Microscope(Zeiss, Germany) (Plain Apochromat 40X/1.2 WI objective, exposure time 25 ms). Images were analyzed using ImageJ software. To measure ROS levels, a circular region of interest was chosen in the zone of the root hair tip cytoplasm. To measure apoplastic ROS, the seedlings were incubated with 50 μM Amplex™ UltraRed Reagent (AUR) (Molecular Probes, Invitrogen) for 15 min in darkness and rinsed with liquid 0.5X MS media (Duchefa, Netherlands). Root hairs were imaged with a Zeiss LSM5 Pascal (Zeiss, Germany)) laser scanning confocal microscope (Excitation:

543 nm argon laser; Emission: 560–610 nm, Plain Apochromat 40X/1.2 WI objective). Quantification of the AUR probing fluorescence signal was restricted to apoplastic spaces at the root hair tip and quantified using the ImageJ software. Fluorescence AU were expressed as the mean ± SD using the GraphPad Prism 8.0.1 (USA) statistical analysis software. Results of both ROS measurements are representative of two independent experiments, each involving 10–15 roots and aproximately, between 10 and 20 root hairs per root were observed.

***In silico*** **analysis**. The *in silico* analysis of *PRX62* and *PRX69* genes expression in the roots were performed using ePlant browser of Araport, Tissue-Specific Root eFP (http://bar.utoronto.ca/eplant/)[33]. *EXPANSIN 7* (*EXP7*) as a RH-specific gene was included for comparison.

**RNA-seq analyses**. Frozen samples were ground in liquid nitrogen and total RNAs were extracted using the RNeasy Plant Mini Kit (Qiagen, Courtaboeuf, France). RNA quantification was performed using a spectrophotometer ND-1000 (NanoDrop, Wilmington, DE, USA) and RNA quality was assessed on an Agilent 2100 Bioanalyzer (Agilent Technologies, Courtaboeuf, France). RNA seq experiments were performed on an Illumina HiSeq 3000 at the GeT-PlaGe platform (get.genotoul.fr, Auzeville, France) according to the standard Illumina protocols. Short pair-end sequencing reads generated were analyzed using the commercial CLC Genomic Workbench 8.0 software (CLC bio, Aarhus, Denmark). This section is adapted from the 3D RNA-seq package output "Results"[74,75] as this was the selected tool to analyze differential expression in our datasets. For the RNA-seq datasets we analyzed 2 datasets, one with 16 factor groups (Col.X10, Col.X22, Bu.X10, Bu.X22, Sf.X10, Sf.X22, Te.X10, Te.X22, Wc.X10, Wc.X22, P62.X10, P62.X22, P69.X10, P69.X22, P6269.X10, P6269.X22) each with two biological replicates (32 samples in total). Quantification of transcripts using salmon quant[76] from Galaxy.org or salmon-1.5.1_linux_x86_64 version in a personal computer. The index of the transcriptome was built using The Arabidopsis Thaliana Reference Transcript Dataset 2[77] (AtRTD2) obtained from https://ics.hutton.ac.uk/atRTD/. For the data pre-processing, read counts and transcript per million reads (TPMs) were generated using tximport R package version 1.10.0 and lengthScaledTPM method[78] with inputs of transcript quantifications from tool salmon[76]. Low expressed transcripts and genes were filtered based on analyzing the data mean-variance trend. The expected decreasing trend between data mean and variance was observed when expressed transcripts were determined as which had $\geq 1$ of the 32 samples with count per million reads (CPM) $\geq 1$, which provided an optimal filter of low expression. A gene was expressed if any of its transcripts with the above criteria was expressed. The TMM method was used to normalize the gene and transcript read counts to $log_2$-CPM[79]. The principal component analysis (PCA) plot showed the RNA-seq data did not have distinct batch effects. For the DE, DAS, and DTU analysis, the voom pipeline of limma R package was used for 3D expression comparison[80,81]. To compare the expression changes between conditions of experimental design, the contrast groups were initially set as Col.X10-Col.X22, Bu.X10-Bu.X22, Sf.X10-Sf.X22, Te.X10-Te.X22, Wc.X10-Wc.X22, P62.X10-P62.X22, P69.X10-P69.X22, P6269.X10-P6269.X22. For DE genes/transcripts, the $log_2$ fold change ($L_2FC$) of gene/transcript abundance were calculated based on contrast groups, and significance of expression changes were determined using t-test. *P*-values of multiple testing were adjusted with BH to correct false discovery rate (FDR)[82]. A gene/transcript was significantly DE in a contrast group if it had adjusted *p*-value < 0.01 and $L_2FC \geq 1.5$. Heatmap: Hierarchical clustering was used to partition the DE genes into 10 clusters with euclidean distance and ward.D clustering algorithm[83]. ComplexHeatmap R package version 1.20.0 was used to make the heat-map.

**Alternative splicing analysis**. At the alternative splicing level, DTU transcripts were determined by comparing the $L_2FC$ of a transcript to the weighted average of $L_2FCs$ (weights were based on their standard deviation) of all remaining transcripts in the same gene. A transcript was determined as significant DTU if it had adjusted *p*-value < 0.01 and $\Delta PS \geq 0.15$. For DAS genes, each individual transcript $L_2FC$ were compared to gene level $L_2FC$, which was calculated as the weighted average of $L_2FCs$ of all transcripts of the gene. Then *p*-values of individual transcript comparisons were summarized to a single gene level *p*-value with F-test. A gene was significantly DAS in a contrast group if it had an adjusted *p*-value < 0.01 and any of its transcript had a Δ Percent spliced (ΔPS) ratio $\geq 0.15$.

**Gene ontology analysis**. Gene ontology (GO) terms assignment for the DE genes datasets were obtained using the PantherDB tool (http://go.pantherdb.org/index.jsp). An enrichment test was performed for the following categories: BP (biological process), MF (molecular function), and CC (cellular component). *p*-values were obtained using the Fisher exact test and corrected for multiple testing using FDR. The enrichment factor (EF) was estimated as the ratio between the proportions of genes associated with a particular GO category present in the dataset under analysis, relative to the number of genes in this category in the whole transcriptome analyzed. We considered the whole transcriptome as those genes that are expressed at least in one of the evaluated conditions. Bubble plots were

generated, using a custom script written in Python language (https://github.com/Lucas-Servi/makeGO), for all those categories for which the adjusted *p*-value (FDR) was lower than 0.01.

**SHAM treatment**. Seeds were germinated on agar plates at 22 °C in a growth chamber in continuous light. After 4 days plants were transferred to agar plates with or without 65 μM of SHAM (Salicylhydroxamic acid; Sigma Aldrich, USA), then grown 3 days at 22 °C followed by 3 days at 10 °C or 6 days at 22 °C (control). Root hair phenotype was measured and confocal microscopy analysis was performed.

**SS-TOM and SS-TOM-Long-EXT constructs**. The binary vector pART27, encoding tdTomato secreted with the secretory signal sequence from tomato polygalacturonase and expressed by the constitutive CaMV 35S promoter (pART-SS-TOM), was a kind gift of Dr. Jocelyn Rose, Cornell University. The entire reporter protein construct was excised from pART-SS-TOM by digesting with *Not*I. The resulting fragments were gel-purified with the QIAquick Gel Extraction Kit and ligated using T4 DNA Ligase (New England Biolabs) into dephosphorylated pBlueScript KS+, also digested with *Not*I and gel-purified, to make pBS-SS-TOM. The plasmid was confirmed by sequencing with primers 35S-FP (5′-CCTTCGCAAGACCCTTCCTC-3′) and OCS-RP (5′-CGTGCACAACAGAATT GAAAGC-3′). The sequence of the EXT domain from *SlPEX1* (NCBI accession AF159296) was synthesized and cloned by GenScript into pUC57 (pUC57-EXT). The plasmid pBS-SS-TOM-Long-EXT was obtained by digesting pUC57-EXT and pBS-SS-TOM with *Nde*I and *Sgr*AI, followed by gel purification of the 2243 bp band from pUC57-EXT and the 5545 bp band from pBS-SS-TOM, and ligation of the two gel-purified fragments. The pBS-SS-TOM-Long-EXT plasmid was confirmed by sequencing with 35S-FP, OCS-RP, and tdt-seq-FP (5′-CCCGTTCAA TTGCCTGGT-3′). Both pBS plasmids were also confirmed by digestion. The binary vector pART-SS-TOM-Long-EXT was made by gel purifying the *Not*I insert fragment from the pBS-SS-TOM Long EXT plasmid and ligating it with pART-SS-TOM backbone that had been digested with NotI, gel purified, and dephosphorylated. This plasmid was confirmed by sequencing. The construct SS-TOM and SS-TOM-Long-EXT where transformed into Arabidopsis plants. The secretory sequence (SS) from tomato polygalacturonase is MVIQRNSILLLIII-FASSISTCRSGT (2.8 kDa) and the EXT-Long domain sequence is BAAAAAAA CTLPSLKNFTFSKNIFESMDETCRPSESKQVKIDGNENCLGGRSEQRTEKECFP VVSKPVDCSKGHCGVSREGQSPKDPPKTVTPPKPSTPTTPKPNPSPPPPKTLP PPPKTSPPPPVHSPPPPPVASPPPPVHSPPPPVASPPPPVHSPPPPPVASPPPPV HSPPPPVASPPPPVHSPPPPVHSPPPPVASPPPPVHSPPPPVHSPPPPVHSPPP PVHSPPPPVHSPPPPVASPPPPVHSPPPPVHSPPPPVHSPPPPVASPPPPVHSP PPPPVASPPPPVHSPPPPVASPPPPVHSPPPPVASPPPPVHSPPPPVHSPPPPV HSPPPPVASPPPALVFSPPPPVHSPPPPAPVMSPPPPPTFEDALPPTLGSLYASPPP PIFQGY* 395–(39.9 kDa). The predicted molecular size for SS-TOM protein is 54.2 kDa and for SS-TOM-EXT-Long Mw is 97.4 kDa.

**Modeling and molecular docking between PRXs and EXTs**. Modeling and molecular docking: cDNA sequences of PRXs were retrieved from TAIR (*PRX62*: AT5G39580 and *PRX69*: AT5G64100) and NCBI Nucleotide DB (PRX24Gv: *Vitis vinifera* peroxidase 24, GvEP1, LOC100254434). Homology modeling was performed for all PRXs using modeler 9.14[84], using the crystal structures 1PA2, 3HDL, 1QO4, and 1HCH as templates, available at the protein data bank. 100 structures where generated for each protein and the best scoring one (according to DOPE score) was picked. The receptor for the docking runs was generated by the prepare_receptor4 script from autodock suite, adding hydrogens and constructing bonds. Peptides based on the sequence PYYSPSPKVYYPPPSSYVYPPPPS were used, replacing proline by hydroxyproline, and/or adding O-Hyp glycosylation with up to four arabinoses per hydroxyproline in the fully glycosylated peptide and a galactose on the serine, as it is usual in plant O-Hyp[85]. Ligand starting structure was generated as the most stable structure by molecular dynamics[6]. All ligand bonds were set to be able to rotate. Docking was performed in two steps, using Autodock vina[86]. First, an exploratory search over the whole protein surface (exhaustiveness 4) was done, followed by a more exhaustive one (exhaustiveness 8), reducing the search space to a $75 \times 75 \times 75$ box centered over the most frequent binding site found in the former run.

**Phylogenetic tree of the Arabidopsis Class III PRXs gene sequences**. DNA sequences from all PRXs genes were retrieved from RedoxiBase[87] (https://peroxibase.toulouse.inra.fr/). Multiple alignment of the sequences was made with MAFFT software[69] (https://mafft.cbrc.jp/alignment/software/). Alignment curation, tree inference, and rendering were made using NGPhylogeny.fr[88] (https://ngphylogeny.fr/). For tree visualization was used iTOL[89] v6.4 (https://itol.embl.de/). Data visualization of tissue-specific expression of PRXs in roots was from ePlant Browser[33] (http://bar.utoronto.ca/eplant/).

**Reporting summary**. Further information on research design is available in the Nature Research Reporting Summary linked to this article.

## Data availability

Data supporting the findings of this work are available within the paper and its Supplementary Information and Source Data files. The datasets and plant materials generated and analyzed during this study are available from the corresponding author J.M.E. upon request. Transcriptome data were deposited in NCBI's Gene Expression Omnibus (GEO) under the project number PRJNA793162. Source data are provided with this paper.

## Code availability

Bubble plots were generated using a custom script written in Python language (https://github.com/Lucas-Servi/makeGO).

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

## Acknowledgements

We would like to thank Margaret Fleming and Patricia Bedinger for sharing SS-TOM-Long-EXT and SS-TOM constructs and Jorge Muschietti for his valuable comments on this work. We thank NASC (Ohio State University) for providing T-DNA lines seed lines. J.M.E., C.M.F., E.P., J.B., A.D.N. and F.A. are investigators of the National Research Council (CONICET) from Argentina. This work was supported by grants from ANPCyT (PICT2017-0066, and PICT2019-0015 to J.M.E.). In addition, this research was also funded by ANID—Programa Iniciativa Científica Milenio ICN17_022, NCN2021_010 and Fondo Nacional de Desarrollo Científico y Tecnológico [1200010] to J.M.E.

## Author contributions

J.M.P. performed most of the experiments, analyzed the data, and wrote the paper. P.R. measured the peroxidase activity and PRXs expression in single *prx* mutants and selected accessions, analyzed the subcellular localization of PRXs, cloned the PRXs, generated the 35S~pro~:PRXs constructs, and produced the RNA-seq data. L.K. performed GWAS measurements. C.M.F. performed GWAS and haplotype analysis. L.S., R.T, C.M. and E.P. performed the bioinformatics analysis. V.B.G., J.M.P., C.B., E.M., D.R.R.G, Y.R. and M.C. analyzed part of the data. J.B provided the accessions and analyzed the data. L.S., E.P. and C.M. analyzed the RNA-seq data. L.F. and F.A analyzed the data and commented on the manuscript. In addition, L.S and E.P. analyzed the alternative splicing. C.D. produced the RNA-seq data and analyzed the results. A.A.A. and A.D.N. performed the molecular modeling/docking analysis. J.M.E. designed research, analyzed the data, supervised the project, and wrote the paper. All authors commented on the results and the manuscript. This manuscript has not been published and is not under consideration for publication elsewhere. All the authors have read the manuscript and have approved this submission.

## Competing interests

The authors declare no competing interests.
