## [Peer Review File · Nature Communications]

Apoplastic class III peroxidases PRX62 and PRX69 promote Arabidopsis root hair growth at low temperatureREVIEWER COMMENTS

Reviewer #1 (Remarks to the Author):

In this work, the authors investigated the molecular basis of the more robust RH growth at low temperatures in Arabidopsis. They use a GWAS approach on Arabidopsis thaliana natural accessions and identify PRX62 as a critical factor responded to this strong RH growth under low-temperature stress. PRX69 was also considered to play a role in this process. Next, the author confirmed the effect of these two proteins on root hair elongation under low temperature through a series of experiments and proposed his model diagram. There are some major concerns describe below:

(1) It is very hard to conclude that PRX62 and PRX69 function in root hair growth at low-temperature. Firstly, based on GWAS under 22°C, the signal of PRX69 was detected but failed to pass the statistic threshold. Secondly, overexpression PRX62 and PRX69 clearly enhanced RH growth under 22°C based on Fig3E. Thirdly, the apoROS level was significantly affected in the double mutant and the two overexpressors under 22°C based on Fig5B.

(2) The statistic meaning was not clear for the description "after filtering SNPs for a 10% minor allele frequency in the 22°C to 10°C RH length GWAS, a leader SNP....." (Line78-79). This filtering should be taken in both conditions for GWAS, but not only one condition.

(3) Four linked SNPs were identified in PXR62 to be associated with the RH length under 10°C, with a nonsynonymous mutation. Then, the author characterized that the gene induction rate by low temperature contributed the phenotypic variation, based on the gene expression patterns of one short RH and two long RH accessions (Fig.2D). It was very difficult to reach the conclusion. At least 30 accessions should be tested for the two major haplotypes based on the four SNPS to test whether the phenotypic variance was resulted from the expression different. If so, this experiment also can infer that the unidentified causal variation in gene expression regulation region brought about the detected significance of the SNPs due the genetic linkage, even though so far the causal variation was not defined yet.

(4) In Fig.5B, it was shown that in prx62 prx69 double mutant, the apoROS level was lower than WT. However, PRX62 and PRX69 are ROS scavenging enzymes and have been shown to be secreted to the apoplastic space. Why did the double mutant have lower apoROS content? On the other hand, Fig.5A showed that in the double mutant, there was a higher cytoROS content than WT. It seemed to be consistent that lacking the ROS scavenging enzyme may resulted a higher ROS level, but PRX62 and 69 considered to function in apoplastic region. It is not easy to accept that the increase of their expression leads to the increase of apoROS.

(5) Another weak point of this work was that the author stated that the RSL4 transcription factor directly controls the expression of PRX69, but the only experimental data was shown in FigS9. A mild enrichment of RSL4 in the second E-box of PRX69 promoter was detected. And mutant of rsl4 and rsl2 failed to change PRX69 expression under 10°C and/or 22°C. Overexpression RSL4 enhanced PRX69 expression

under 22°C but not 10°C. RLS4 did not bind to PRX62 promoter at all; however, in the *rsl4 rsl2* mutant the PRX62 expression was greatly reduced under 10°C. Together, the data were rather confusing.

(6) The authors use much content to describe the splice variants of PRX62 and PRX69. The two full-length transcripts are the primary transcript, and the two short transcripts do not seem necessary.

(7) There were some mistakes in the manuscript. The author should pay attention to them. For example, 106 *Arabidopsis thaliana* accessions were used in this study, but there were 110 in the haplotypes classification. Please confirm. There was no Fig.4C, but it was cited in Line 171. The format of citation in the context was not consistent such as in Line 24-30.

Reviewer #2 (Remarks to the Author):

This work reports novel function of *Arabidopsis* PRX62 and PRX69 involved in the cold dependent-root hair growth through GWAS, transcriptome and molecular and biochemical analyses. Overall data support the significant role of PRX62 in the cold induced root hair elongation. However, function of PRX69 coupled with PRX2 is not very clear. Logic of selecting PRX69 is weak. Authors may need additional explanation through phylogenetic analysis of PRX family.

GWAS analysis suggests the significance of PRX62 in this report but I am curious about possibility of additional candidate genes and if authors change the criteria to select candidate gene from GWAS, more candidate genes can be suggested. In addition, phenotype changes among PRX62 haplotypes can not fully explain the root hair elongation by cold in the GWAS analysis. Authors need to provide the answer for this possibility.

Root hair change in cold is regulated by not only PRX62 but also others based on the data in Figure S1 showing overall changes of diverse varieties and Figure S3 showing diversity among PRX62 haplotypes. In the others, PRX69 might be included but other members may have more important roles.

In Figure S7, cold induced root hair elongation is still observed at *prx62prx69* mutant, indicating that there are additional PRX members involved in this process. Authors need to suggest and define the roles of additionally required members. Regarding this, transcriptome analysis revealed several PRX genes

showing upregulation in the double mutant and those PRXs might be good candidates. In Figure 4, PRX genes in cluster 4 can be potential candidates or if there are PRX genes in clusters 6 and 10, they are the better candidate to compensate the defects of PRX62 and PRX69.

Authors need to clarify the benefit of enhancing ROS homeostasis under cold stress. For example, all species with longer root hair under cold stress as well as PRX62 overexpression lines should be linked with cold stress tolerance. To expand this finding to other crop plants, authors need to provide the evidence that crop plants also have positive relationship between cold (cold stress tolerance) and root hair elongation.

In Figure S9, PRX62 expression is not dependent on the expression of RSL4 based on the data in *rsl4* mutant and *RSL4* ox under cold condition. PRX69 expression showed dosage dependent expression by RSL4 under normal condition but downregulated under cold condition. Current data does not well support the regulation of PRX69 genes by RSL4 for the root hair elongation under cold stress. Authors need more clear evidence to support the model in Figure 6. Alternatively, authors can change the model based on revised content.

In addition, regulation of PRX69 by RSL4 is not very obvious compared to those with EXP7 and LRX1. Authors may need to tone down the description on the regulation.

There are mismatches between text and figure legend in figure 4. Please correct it.

In the figure file, some characters are bold. Is there any reason? If so, please add the explanation in the figure legend.

Reviewer #3 (Remarks to the Author):

This paper reports interesting findings on root hair low temperature responses, the importance of reactive oxygen homeostasis for root hair cell expansion; and two peroxidases of particular significance are identified.

This reviewer identifies as a plant biochemist and the text referring to figure 1 as well as its legend was really hard to read. I would be happy to go directly from (line 81):This SNP is located in the intron of PEROXIDASE62 (PRX62, AT5G39580) to (line 92) the full-length transcript of PRX62 (AT5G39580.1) is detectable in (the) Col-0 (that) and it increased up to 2.54 log₂FC in roots under low-temperature (RNA-seq). This was further

confirmed by RT-qPCR (Figure S4).

What is in between, line 82-91, does not seem to be part of nailing PRX62 but makes the argument unnecessarily hard to follow. I suspect that it is interesting in its own right for the molecular geneticist. If so, move it, and the associated elements of figure 1, to supplemental.

The manuscript suffers from a lack of discussion of, or discrimination between maintaining ROS homeostasis and EXT cross-linking. Given the number of co-regulated PRXs (Figure 4A) it is plausible that PRX62 and PRX69 play roles in ROS homeostasis while some of the co-expressed PRXs catalyse EXT cross-linking. Interestingly, and corroborating Figure 4A, is an overlap of five PRXs between Figure 4A and table 5 in DOI: 10.1039/c9np00028c. The number of PRXs in both Figure 4A and this table 5 is small enough (out of 73) that this is hardly just by chance. A few sentences of discussion and a citation would be relevant.

What is actually meant by ROS homeostasis and plausible mechanisms to regulate it also deserve a few words, not in the least because these secreted PRXs seem to affect cytosolic rather than apoplastic ROS levels.

In short, the concluding statement that these PRXs are directly involved in EXT cross-linking is too quick.

Stephen Fry has pioneered the idea of ROS involvement in cell expansion, see DOI: 10.1104/pp.109.139204. He deserves citation (not necessarily of this paper. It is but an example among several).

apo/cyt is easy to guess but an undefined abbreviation should not appear in the Abstract.

Materials and Methods should have a Plant Material section in which Rd-0, Te-0 etc are defined and how a reader interested in acquiring them (and the rest of the 106) can get them.

line 106, enhance is not a noun. enhance in -> enhancement of ...

Some literature references are a mixture of author (year) and by number.

Reviewer #4 (Remarks to the Author):

This manuscript describes the involvement of two apoplastic peroxidases, PRX62 and PRX69, in the regulation of root hair growth at moderate-low temperature in *Arabidopsis thaliana*. First, the root hair growth was analysed in a very large set of *Arabidopsis* accession lines, and moderate-low temperature was shown to induce the enhancement of root hair growth in all these lines. A Genome-Wide Association Studies was performed to identify the actors in this response and the PEROXIDASE62 gene appeared as a strong candidate. While looking for other PRX genes, the authors also identified the PEROXIDASE69 gene to be strongly expressed in root hair and slightly induced at moderate-low temperature. The implication of these two genes in root hair growth at moderate-low temperature was then investigated using corresponding single and double mutants as well as overexpression constructs. The authors demonstrate that PRX62 and PRX69 participate in the regulation of root hair growth at moderate-low temperature although PRX69's role remains less clear. They also provide insights regarding the function of PRX62 and PRX69, and propose a model in which these two peroxidases would modulate the ROS level and the rate of extensin cross-linkings, and would ultimately regulate root hair elongation in moderate-low temperature.

The manuscript is very pleasant to read and present quite interesting and novel findings with well-designed experiments. However, I think the manuscript could benefit from some clarifications.

The phenotyping of root hair cell growth in *Arabidopsis* accession lines represents quite a big screening and provide very interesting information regarding the root hair growth upon low-moderate temperature. I am a bit surprised that this is only presented as supplemental and not as a main figure for it nicely introduces the manuscript.

Overall, the implication of PRX69 appears less obvious than PRX62. If the expression in root hair is quite high, the upregulation at 10°C is mild (Fig 2D). In addition, the constructs overexpressing PRX69 do not display any difference in root hair growth compared to WT (Fig. 3E). These data may need to be discussed in the manuscript and/or the conclusions regarding PRX69 may be a bit more moderate.

The analysis of the mutants and the overexpressing constructs is comprehensive. The authors could indicate where the T-DNA is inserted in the mutants and show the insertion sites in either Fig 2 or S6.

The complementation of the *prx62prx69* double mutant observed only with 35SproPRX62 and not with 35SproPRX69 is intriguing, especially considering that *prx62* single mutant does not show any phenotype. How do the authors explain this?

The statistical comparisons and analyses need to be clarified. In the experimental procedures, it is stated that for root hair phenotype 10 fully elongated root hair were measured from 15-20 roots, but length values were reported as the mean of 3 replicates. This is confusing to me; I would expect that reported length values would be a mean of 15-20 replicates. Or does it mean that the 10 fully elongated root hair were measured from 15-20 roots in three independent experiments?

Also, when showing statistical significance, it is important to state to what reference the comparison was performed, especially in Figure 3D and Figure 5. In Figure 3D, was the comparison made versus the "Col-0, 22°C" condition? Then, what about the difference between 22°C and 10°C in each mutant, is the increase of peroxidase activity in *prx62prx69* double mutant significant? This increase (10°C vs 22°C) looks very mild compared to the one observed in Col-0. This result is of interest and may deserve to be discussed considering this manuscript describes the implication of PRX62 and PRX69 in root hair elongation in response to moderate-low temperature. In Figure 5, each graph seems to display two degrees of comparison but the reference for each degree of comparison is not stated. In Figure 5C, I could not determine what were the statistical comparisons.

Given the large number of peroxidases, it is slightly surprising to see that peroxidase activity appears to be more strongly reduced when PRX62 and PRX69 are mutated than when the PRX inhibitor SHAM is used (Fig 2C and 3D). Do the authors have any comment on this? Was the peroxidase activity also assayed in the mutants treated with SHAM? Could the effects of PRX62-69 mutations and SHAM treatment be additional? Not that I think this experiment is absolutely required here but I'm curious.

The authors have identified several genes coding for extensins that are upregulated in lower temperature in Col-0 and to a lesser extent in the *prx62prx69* double mutant (Figure 4A). I wonder if this is also the case for genes coding for the enzymes involved in extensin glycosylation (e.g. SGT1, HPAT1-3, XEG113 etc...). One can expect that increased level of extensins would be reflected in higher expression of these genes.

The strong decrease of signal from the extensin reporter after SHAM treatment is very interesting although it might have been even more informative if the extensin reporter was used in the *prx62prx69*

double mutant. That being said, I can imagine that SHAM treatment was faster and more straightforward to perform.

Small corrections:

Line 24: The references “Valerio et al. 2004” and “Passardi et al. 2004a” need to be formatted.

Line 111: Replace “were positively correlated” by “coincided”, no math correlation presented.

Line 130: The reference “Jemmat et al. 2020” needs to be formatted.

Lines 169 and 171: Figure 4A instead of Figure 4A-B and Figure 4B instead of Figure 4C.

Figure 4 contains only A and B.

Fig 1A: Why not show Bu-2 since the accession is later used for PRX62 and PRX69 expression (Fig 2D and Fig 4C)?

Maybe better put “22°C->10°C long RH” on top of Col-0 in Fig 1A. The legend should also mention that seedlings were transferred to 10°C and not “grown”.

REVIEWER COMMENTS

Reviewer #1 (Remarks to the Author):

In this work, the authors investigated the molecular basis of the more robust RH growth at low temperatures in Arabidopsis. They use a GWAS approach on Arabidopsis thaliana natural accessions and identify PRX62 as a critical factor responded to this strong RH growth under low-temperature stress. PRX69 was also considered to play a role in this process. Next, the author confirmed the effect of these two proteins on root hair elongation under low temperature through a series of experiments and proposed his model diagram. There are some major concerns describe below:

(1) It is very hard to conclude that PRX62 and PRX69 function in root hair growth at low-temperature. Firstly, based on GWAS under 22°C the signal of PRX69 was detected but failed to pass the statistic threshold.

R/ In the manuscript, we described a GWAS approach to identify loci involved in the regulation of root hair growth. We phenotyped the population in two conditions: 1) at 22° and 2) at 22°->10°. Only one statistically significant association was found in the GWAS for 22°->10° at Chromosome 5 (SNP position 15847854, TAIR10) within the locus *PRX62*. No associated SNPs were found within the genomic region of *PRX69* either at 22° or at 22°->10°. As the reviewer pointed out, a weak peak is seen at the region harboring *PRX62* in the GWAS for 22°. This peak does not surpass the statistical threshold and therefore was not considered for further analysis. This result can be explained because there is a lower variation for RH length at 22° compared to the variation observed at 10° (CV 39% vs. 51%, respectively). Thus, this lower variation is not sufficient to be explained by polymorphisms present in the *PRX62* locus.

We cannot discard the possibility that for GWAS at 22°C, the variation observed in the assayed population is lower than the variation actually present in the ideal population. However, against this hypothesis, we do detect significantly associated SNPs in GWAS at 22°C->10°C, meaning that the population was enough both in the quantity of individuals and in diversity to perform GWAS.

Secondly, overexpression PRX62 and PRX69 clearly enhanced RH growth under 22°C based on Fig3E.

R/ This is not contradictory at all with our hypothesis and matches the fact that at 10°C the expression of *PRX62* is upregulated. Overexpressing *PRX62* is simulating a 10°C scenario in a constitutive manner, promoting RH growth even when the low temperature is not applied. Please note that the response of RH to lower temperature is still seen in the overexpression lines.

Thirdly, the apoROS level was significantly affected in the double mutant and the two overexpressors under 22°C based on Fig5B.

R/ We based our conclusion on the double mutant *prx62 prx69* phenotype but clearly *PRX62* plays a more predominant role based on the complementation assays on the double *prx* mutant by overexpression (35S-lines) as well as the overexpression in the Wt Col-0.

(2) The statistic meaning was not clear for the description "after filtering SNPs for a 10% minor allele frequency in the 22°C to 10°C RH length GWAS, a leader SNP....." (Line 78-79). This filtering should be taken in both conditions for GWAS, but not only one condition.

R/ Filtering of SNPs for a minor allele frequency can be done before or after running GWAS. In any case, it is of interest to do it when GWAS results showed significant associations, as taking a threshold of minor allele frequency enables the identification of fairly well represented alleles in the population of study. As we haven't found SNP significantly associated at 22°C, we did not perform any further corrections for minor allele frequency on this data set. Given the fact that this procedure could lead to a misunderstanding, we applied MAF corrections to both experiments. Applying the MAF correction to the GWAS at 22°C did not result in any further changes. We clarified that in the text.

(3) Four linked SNPs were identified in *PRX62* to be associated with the RH length under 10°C with a non-synonymous mutation. Then, the author characterized that the gene induction rate by low temperature contributed to the phenotypic variation, based on the gene expression patterns of one short RH and two long RH accessions (Fig. 2D). It was very difficult to reach the conclusion. At least 30 accessions should be tested for the two major haplotypes based on the four SNPs to test whether the phenotypic variance was resulted from the expression difference. If so, this experiment also can infer that the unidentified causal variation in gene expression regulation region brought about the detected significance of the SNPs due the genetic linkage, even though so far the causal variation was not defined yet.

R/ From our GWAS results, these four SNPs are linked and define two major (the most common) and opposite haplotypes found in the population. Accessions carrying these two haplotypes showed a significantly different RH length at 22°C to 10°C. These differences agree with the allelic effect of the highest-LOD SNP found in GWAS at position 15847854.

The reviewer is completely right, from our analysis, we cannot discard that there are other SNPs linked to these four-SNP haplotypes contributing to the overall variation and we certainly do not exclude the possibility that the causal variation is located at the promoter of *PRX62* locus. This is the main principle of doing GWAS, we can still detect significant associations because of SNPs that are scored and linked to the causal SNPs (not scored). In our GWAS, SNPs in the promoter region of *PRX62* were not scored. To test whether the causal variation might be in the promoter region of *PRX62*, we mined 1001 Arabidopsis genomes and downloaded *PRX62* genomic sequences including ca. 2000 bp upstream the start site and calculated the linkage disequilibrium for the complete region. Our goal was to check whether high-LOD SNPs in the coding region are linked to SNPs, not previously scored, in the promoter region. In fact, we identified that at least four SNPs in the promoter region of *PRX62* (at 35, 101, 417 and 489 bp upstream the ATG) are in significant linkage disequilibrium with the high-LOD SNPs ($r^2 > 0.1$, $p < < 0.001$). Polymorphisms at these positions or at positions in LD with them could explain the RH-length variation at 10°C and the different expression levels of *PRX62*. We add this data in **Figure S4**.

(4) In Fig. 5B, it was shown that in *prx62 prx69* double mutant, the apoROS level was lower than WT. However, *PRX62* and *PRX69* are ROS scavenging enzymes and have been shown to be secreted to the apoplast space. Why did the double mutant have lower apoROS content? On the other hand, Fig. 5A

showed that in the double mutant, there was a higher cytoROS content than WT. It seemed to be consistent that lacking the ROS scavenging enzyme may result in a higher ROS level, but PRX62 and 69 are considered to function in the apoplastic region. It is not easy to accept that the increase of their expression leads to the increase of apoROS.

R/ Class III PRXs are not only ROS scavengers. Mechanistically, Class III PRXs can reduce the available apoplastic H₂O₂ (via the peroxidative cycle) using various electron donors (e.g., EXT tyrosines, monolignols, ferulic acids, and suberin) leading to crosslinking of EXT Tyr-containing domains to form EXT dimers and/or trimers. Extracellular anion superoxide is generally produced by plasma membrane NADPH oxidase/respiratory burst oxidase homolog (RBOH) proteins, including the respiratory burst oxidase homolog C, which use intracellular NADPH to reduce extracellular O₂ to form peroxides in the apoplast (Passardi *et al.*, 2004, 2005; Held *et al.* 2004; O'Brien *et al.*, 2012; Lüthje *et al.*, 2013; Mignolet-Spruyt *et al.*, 2016; Mishler-Elmore *et al.* 2021). Additionally, Class III PRXs can also produce H₂O₂ and subsequently OH⁻ radicals (via the hydroxylic cycle) to regulate extracellular ROS production (Passardi *et al.*, 2004). Hence, both cycles may regulate H₂O₂ production and cytoplasm/apoplast ROS homeostasis. This was clearly explained in the text.

(5) Another weak point of this work was that the author stated that the RSL4 transcription factor directly controls the expression of PRX69, but the only experimental data was shown in FigS9. A mild enrichment of RSL4 in the second E-box of PRX69 promoter was detected. And mutant of *rsl4* and *rsl2* failed to change PRX69 expression under 10°C and/or 22°C. Overexpression of RSL4 enhanced PRX69 expression under 22°C but not 10°C. RSL4 did not bind to PRX62 promoter at all; however, in the *rsl4 rsl2* mutant the PRX62 expression was greatly reduced under 10°C. Together, the data were rather confusing.

R/ We do agree that these results require further investigations. For this reason, we decided to remove these results from this updated version of the manuscript.

(6) The authors use much content to describe the splice variants of PRX62 and PRX69. The two full-length transcripts are the primary transcripts, and the two short transcripts do not seem necessary.

R/ This part was described but trimmed as suggested by this reviewer.

(7) There were some mistakes in the manuscript. The author should pay attention to them. For example, 106 *Arabidopsis thaliana* accessions were used in this study, but there were 110 in the haplotypes classification. Please confirm. There was no Fig.4C, but it was cited in Line 171. The format of citation in the context was not consistent such as in Line 24-30.

R/ The manuscript was carefully proofread. We thank this reviewer for all the suggestions.

In **Table S1** there are 107 lines described. I carefully checked the haplotype classification and we then got 80 with CTGT, 16 with TGAA, 4 with TGGA, 4 with CTGA, this makes 104. The remaining 3 lines were not included in the haplotype plot because the haplotypes were unique (TTGT, CGGT, CGGA).

Reviewer #2 (Remarks to the Author):

This work reports novel function of Arabidopsis PRX62 and PRX69 involved in the cold dependent-root hair growth through GWAS, transcriptome and molecular and biochemical analyses. Overall data support the significant role of PRX62 in the cold induced root hair elongation. However, function of PRX69 coupled with PRX2 is not very clear. Logic of selecting PRX69 is weak. Authors may need additional explanation through phylogenetic analysis of PRX family.

R/ We have added a new Supplementary Figure (**Figure S7**) where we present the Phylogenetic tree of the 75 apoplasmic class III PRXs genes (73 genes and 2 pseudogenes) in *Arabidopsis* and PRX62 and PRX69 cluster in the same subclade (**A**) with an identity percentage 57.2% between the PRX62 and PRX69 aminoacidic sequences (**B**). In addition, both PRXs PRX62 and PRX69 are highly expressed in root hairs while the other six PRXs of the same subclade do not (results not shown).

A

B

sp Q96511 PER69_ARATH	--MCGYNLELVTFVLVAAVTAQGNRGSNSGGRRPHVDFGNRERVLSVWSSVQ	58
sp Q9FKA4 PER62_ARATH	MGLVLSFALVIFLSCILAV-----YQGTRIEFTTTPAATLVTTTA	46
sp Q96511 PER69_ARATH	SVRSIIANFSLERMFHCEHSGDGSVLLAANTBERTVFRSRFEVIEEAAAR	118
sp Q9FKA4 PER62_ARATH	SPFGIDKVAEELRMNHCEVCSGDGSVLLSFPNERTGAVNHFEEVIEDAERQ	106
sp Q96511 PER69_ARATH	SKACDRTVSCADITLAARDAVVINGSRNEVLERLDGRISQSSDQ--NIPGSDSVKQ	177
sp Q9FKA4 PER62_ARATH	FAKHGVVSCADILAAARDSSVINGSRNEVLERLDGRISQSSDQ--NIPGSDSVKQ	166
sp Q96511 PER69_ARATH	KQDAKKTNTLDVLTWV--SHTIGTAGGLVLRGIFVFNFTGQPFSSIPSVLILAQ	236
sp Q9FKA4 PER62_ARATH	QRKSSIFRINTRDNTLWVGHHTIGTAGGLFITNIFSSN-TAQTMTQTVLQLQL	225
sp Q96511 PER69_ARATH	SPONGG--TRVDEEESVVDKFDTSFLRKVTSSVVLQSLVLRKDETRATIERLLGLRR	294
sp Q9FKA4 PER62_ARATH	SPONGDGSARVDDTSSGNTFDTSYFINLSRNLGTQSHHNTSAAHSLVQEFMA---	282
sp Q96511 PER69_ARATH	SLRFGTEKGMVKMSLIEVTEGSDGETRRCVSIIN	331
sp Q9FKA4 PER62_ARATH	IRGNVQVARSVKMNLGVKVTNGETRRCVSIIN	319

GWAS analysis suggests the significance of PRX62 in this report but I am curious about possibility of additional candidate genes and if authors change the criteria to select candidate gene from GWAS, more candidate genes can be suggested. In addition, phenotype changes among PRX62 haplotypes can not fully explain the root hair elongation by cold in the GWAS analysis. Authors need to provide the answer for this possibility.

R/ The primary criterion to select candidate genes involved in RH growth was the threshold given by the FDR correction for multiple comparisons. For the GWAS 22°C->10°C only one additional SNP surpassed this criterion and it is located at 500 Kb of PRX62 lead-SNP (SNP position 16341915, TAIR10). This additional statistically significant SNP located within *loci* AT5G40810 (Cytochrome C1 family protein) and AT5G40820 (*ATRAD3* protein kinase involved in responses to DNA damage and aluminum tolerance). Regarding GWAS for 22°C we could not find any SNP surpassing our criterion of selection.

The association found for PRX62 with RH length at 10°C explained 21% of the variance for this trait. Hence, as with many other complex traits, there are other loci involved in RH length variation that failed to be detected by GWAS. This could be in part due to a lower effect of those QTL, background variance heterogeneity, or epistasis. In addition, loci that are on-off are not likely to be detected by GWAS as they have been fixed in the course of evolution. There is a clarification added in the manuscript.

Root hair change in cold is regulated by not only PRX62 but also others based on the data in Figure S1 showing overall changes of diverse varieties and Figure S3 showing diversity among PRX62 haplotypes. In the others, PRX69 might be included but other members may have more important roles.

In Figure S7, cold induced root hair elongation is still observed at *prx62prx69* mutant, indicating that there are additional PRX members involved in this process. Authors need to suggest and define the roles of additionally required members.

R/ we do agree with this reviewer on this. For that reason we have performed a new experiment (**Figure S9**) where we used IC_{50} SHAM (65 μ M) in the double mutant *prx62 prx69* to test if there is residual peroxidase activity arising from other PRXs that might act on RH growth specifically at 10°C. As expected, when SHAM was applied in the double mutant *prx62 prx69* background, a strong inhibition is detected suggesting that other PRXs apart from PRX62 and PRX69, might be also important in RH growth at 10°C. This is an interesting aspect to be studied in the future but it is clearly out of the scope of the present work.

Regarding this, transcriptome analysis revealed several PRX genes showing upregulation in the double mutant and those PRXs might be good candidates. In Figure 4, PRX genes in cluster 4 can be potential candidates or if there are PRX genes in clusters 6 and 10, they are the better candidate to compensate the defects of PRX62 and PRX69.

R/ We did not detect other Class III apoplastic PRXs in the remaining clusters in the heat map of the RNA-seq analysis.

Authors need to clarify the benefit of enhancing ROS homeostasis under cold stress. For example, all species with longer root hair under cold stress as well as PRX62 overexpression lines should be linked with cold stress tolerance. To expand this finding to other crop plants, authors need to provide the evidence that crop plants also have positive relationship between cold (cold stress tolerance) and root hair elongation.

R/ In this study and in two previous and very recent studies from our lab (Moison et al. 2021 Molecular Plant; Martinez Pacheco et al. 2021 Plant Signaling & Behavior) we have found that low temperature at 10°C triggers enhanced root hair growth in order to allow root hair to reach those nutrients which mobility is greatly reduced by the effect of the temperature. Probably part of this enhanced growth mechanism is based on the more dynamic ROS homeostasis as well as many other growth indicators such as Ca²⁺, pH etc. We are starting to study these relationships between cold stress tolerance and root hair growth in crops species but they are clearly out of the scope of the present study.

In Figure S9, PRX62 expression is not dependent on the expression of RSL4 based on the data in rsl4 mutant and RSL4 ox under cold condition. PRX69 expression showed dosage dependent expression by RSL4 under normal condition but downregulated under cold condition. Current data does not well support the regulation of PRX69 genes by RSL4 for the root hair elongation under cold stress. Authors need more clear evidence to support the model in Figure 6. Alternatively, authors can change the model based on revised content. In addition, regulation of PRX69 by RSL4 is not very obvious compared to those with EXP7 and LRX1. Authors may need tone down the description on the regulation.

R/ We do completely agree with the reviewers on this. These results require further investigations. For this reason, we decided to remove this part from this updated version of the manuscript.

There are mismatches between text and figure legend in figure 4. Please correct it.

In the figure file, some characters are bold. Is there any reason? If so, please add the explanation in the figure legend.

R/ The manuscript was carefully proofread. We thank this reviewer for all the suggestions.

Reviewer #3 (Remarks to the Author):

This paper reports interesting findings on root hair low temperature responses, the importance of reactive oxygen homeostasis for root hair cell expansion; and two peroxidases of particular significance are identified.

This reviewer identifies as a plant biochemist and the text referring to figure 1 as well as its legend was really hard to read. I would be happy to go directly from (line 81):This SNP is located in the intron of PEROXIDASE62 (PRX62, AT5G39580) to (line 92) the full-length transcript of PRX62 (AT5G39580.1) is detectable in (the) Col-0 (that) and it increased up to 2.54 log₂FC in roots under low-temperature (RNA-seq). This was further confirmed by RT-qPCR (Figure S4). What is in between, line 82-91, does not seem to be part of nailing PRX62 but makes the argument unnecessarily hard to follow. I suspect that it is interesting in its own right for the molecular geneticist. If so, move it, and the associated elements of figure 1, to supplemental.

R/ This part was described but trimmed and better explained as suggested by this reviewer.

The manuscript suffers from a lack of discussion of, or discrimination between maintaining ROS homeostasis and EXT cross-linking. Given the number of co-regulated PRXs (Figure 4A) it is plausible that PRX62 and PRX69 play roles in ROS homeostasis while some of the co-expressed PRXs catalyse EXT cross-linking. Interestingly, and corroborating Figure 4A, is an overlap of five PRXs between Figure 4A and table 5 in DOI: 10.1039/c9np00028c. The number of PRXs in both Figure 4A and this table 5 is small enough (out of 73) that this is hardly just by chance. A few sentences of discussion and a citation would be relevant.

R/ We have added a short discussion about these PRXs.

What is actually meant by ROS homeostasis and plausible mechanisms to regulate it also deserve a few words, not in the least because these secreted PRXs seem to affect cytosolic rather than apoplastic ROS levels.

R/ we do agree with this reviewer that the concept of ROS homeostasis in the root hair tip involves not only apo-ROS but also cyt-ROS. This was the reason we wanted to measure in all lines both types of ROS. This is really an exception in all the publications linked to ROS in plant cells. We have few lines to describe this complex aspect in the text.

In short, the concluding statement that these PRXs are directly involved in EXT cross-linking is too quick.

R/ We have added an analysis of the expression of the SS-EXT-LONG TOM reporter (and the control SS-TOM) in the root hair tip cell walls in the double mutant *prx62 prx69* in comparison to the same reporters in the non-treated Wt Col-0 and SHAM treated Wt Col-0. This is an important new result. In addition, we have performed a homology modeling of PRX62 and PRX69 proteins with GvEP1, an Extensin peroxidase that is able to crosslink extensins *in vitro* and then by *docking* analysis, we obtained interaction energies of these PRXs with extensins substrates. The results obtained further support the function of PRX62 and

PRX69 in extensin interactions and possibly in its crosslinking and insolubilization in the plant cell walls. See **Figure 13**. I think now in this updated version of the manuscript there are stronger evidences on the function of PRX62 and PRX69 linked to the insolubilization of EXTs in the plant cell walls. Still, a direct link to the crosslinking of EXT is missing but this is technically challenging and out of the scope of this work. This will be studied in detail in the near future by our group.

Stephen Fry has pioneered the idea of ROS involvement in cell expansion, see DOI: 10.1104/pp.109.139204. He deserves citation (not necessarily of this paper. It is but an example among several).

R/ We do agree on this with this reviewer. We have added a citation in the introduction to acknowledge Stephen Fry contribution to ROS in the cell wall expansion process.

apo/cyt is easy to guess but an undefined abbreviation should not appear in the Abstract.

R/ They are now defined in the Abstract

Materials and Methods should have a Plant Material section in which Rd-0, Te-0 etc are defined and how a reader interested in acquiring them (and the rest of the 106) can get them.

R/ This was now added in the material and method section.

line 106, enhance is not a noun. enhance in -> enhancement of ...

Some literature references are a mixture of author (year) and by number.

R/ The manuscript was carefully proofread. We thank this reviewer for all the suggestions.

Reviewer #4 (Remarks to the Author):

This manuscript describes the involvement of two apoplastic peroxidases, PRX62 and PRX69, in the regulation of root hair growth at moderate-low temperature in *Arabidopsis thaliana*. First, the root hair growth was analysed in a very large set of *Arabidopsis* accession lines, and moderate-low temperature was shown to induce the enhancement of root hair growth in all these lines. A Genome-Wide Association Studies was performed to identify the actors in this response and the PEROXIDASE62 gene appeared as a strong candidate. While looking for other PRX genes, the authors also identified the PEROXIDASE69 gene to be strongly expressed in root hair and slightly induced at moderate-low temperature. The implication of these two genes in root hair growth at moderate-low temperature was then investigated using corresponding single and double mutants as well as overexpression constructs. The authors demonstrate that PRX62 and PRX69 participate in the regulation of root hair growth at moderate-low temperature although PRX69's role remains less clear. They also provide insights regarding the function of PRX62 and PRX69, and propose a model in which these two peroxidases would modulate the ROS level and the rate of extensin cross-linkings, and would ultimately regulate root hair elongation in moderate-low temperature.

The manuscript is very pleasant to read and present quite interesting and novel findings with well-designed experiments.

R/ We thank the Reviewer for his/her positive appreciation of our work.

However, I think the manuscript could benefit from some clarifications.

The phenotyping of root hair cell growth in *Arabidopsis* accession lines represents quite a big screening and provide very interesting information regarding the root hair growth upon low-moderate temperature. I am a bit surprised that this is only presented as supplemental and not as a main figure for it nicely introduces the manuscript.

R/ Although we do agree with this reviewer on this point, we prefer to keep this phenotypic information as a Supplementary Figure to not overload the current main figures of the manuscript.

Overall, the implication of PRX69 appears less obvious than PRX62. If the expression in root hair is quite high, the upregulation at 10°C is mild (Fig 2D). In addition, the constructs overexpressing PRX69 do not display any difference in root hair growth compared to WT (Fig. 3E). These data may need to be discussed in the manuscript and/or the conclusions regarding PRX69 may be a bit more moderate.

The analysis of the mutants and the overexpressing constructs is comprehensive. The authors could indicate where the T-DNA is inserted in the mutants and show the insertion sites in either Fig 2 or S6.

R/ Please you can check below the information about the T-DNA insertions on these mutants. For further information you can read Jemmat *et al* 2020 (cited on the manuscript).

The complementation of the *prx62prx69* double mutant observed only with 35SproPRX62 and not with 35SproPRX69 is intriguing, especially considering that *prx62* single mutant does not show any phenotype. How do the authors explain this?

The statistical comparisons and analyses need to be clarified. In the experimental procedures, it is stated that for root hair phenotype 10 fully elongated root hair were measured from 15-20 roots, but length values were reported as the mean of 3 replicates. This is confusing to me; I would expect that reported length values would be a mean of 15-20 replicates. Or does it mean that the 10 fully elongated root hair were measured from 15-20 roots in three independent experiments?

R/ The 10 fully elongated root hairs were measured from 15-20 roots in three independent experiments showing a similar statistical behavior. This was corrected in the manuscript.

Also, when showing statistical significance, it is important to state to what reference the comparison was performed, especially in Figure 3D and Figure 5. In Figure 3D, was the comparison made versus the "Col-0, 22°C" condition? Then, what about the difference between 22°C and 10°C in each mutant, is the increase of peroxidase activity in *prx62prx69* double mutant significant? This increase (10°C vs 22°C) looks very mild compared to the one observed in Col-0. This result is of interest and may deserve to be discussed considering this manuscript describes the implication of PRX62 and PRX69 in root hair elongation in response to moderate-low temperature. In Figure 5, each graph seems to display two degrees of comparison but the reference for each degree of comparison is not stated. In Figure 5C, I could not determine what were the statistical comparisons.

Given the large number of peroxidases, it is slightly surprising to see that peroxidase activity appears to be more strongly reduced when PRX62 and PRX69 are mutated than when the PRX inhibitor SHAM is used (Fig 2C and 3D). Do the authors have any comment on this? Was the peroxidase activity also assayed in the mutants treated with SHAM? Could the effects of PRX62-69 mutations and SHAM treatment be additional? Not that I think this experiment is absolutely required here but I'm curious.

R/ The experiment of the root phenotype present in the *prx62prx69* double mutant treated with SHAM was done as recommended and added to the manuscript. This clearly indicates that there is still a remaining peroxidase activity coming from other unknown PRXs when roots are treated at low temp.

The authors have identified several genes coding for extensins that are upregulated in lower temperature in Col-0 and to a lesser extent in the *prx62prx69* double mutant (Figure 4A). I wonder if this is also the case for genes coding for the enzymes involved in extensin glycosylation (e.g. SGT1, HPAT1-3, XEG113 etc...). One can expect that increased level of extensins would be reflected in higher expression of these genes.

R/ We have specifically search for the level of expression of those enzymes acting on the posttranslational modifications of EXTs. Any of these genes seems to be positively regulated by low-temperature (only RRA1) but some of them are downregulated in the double mutant *prx62 prx69*. Here are de plots.

The strong decrease of signal from the extensin reporter after SHAM treatment is very interesting although it might have been even more informative if the extensin reporter was used in the *prx62prx69* double mutant. That being said, I can imagine that SHAM treatment was faster and more straightforward to perform.

R/ **Figure 4D**. A new experiment involving confocal imaging and the measurements of tdTOMATO signal in the root hair tip of the lines: *SS-TOM/prx62prx69* and *SS-TOM-EXT LONG/prx62prx69*, were added. Results were compatible with the previous ones on the SHAM treatment (**Figure 4B-C**).

Small corrections:

Line 24: The references “Valerio et al. 2004” and “Passardi et al. 2004a” need to be formatted.

Line 111: Replace “were positively correlated” by “coincided”, no math correlation presented.

Line 130: The reference “Jemmat et al. 2020” needs to be formatted.

Lines 169 and 171: Figure 4A instead of Figure 4A-B and Figure 4B instead of Figure 4C.

Figure 4 contains only A and B.

Fig 1A: Why not show Bu-2 since the accession is later used for PRX62 and PRX69 expression (Fig 2D and Fig 4C)?

Maybe better put “22°C->10°C long RH” on top of Col-0 in Fig 1A. The legend should also mention that seedlings were transferred to 10°C and not “grown”.

R/ The manuscript was carefully proofread. We thank this reviewer for all the suggestions.

REVIEWERS' COMMENTS

Reviewer #1 (Remarks to the Author):

The authors responded to my concerns and revised the manuscript carefully.

I agree with that PRX62 and PRX69 play roles in root hair (RH) growth at low-temperature, however, I am wondering that they function in RH growth under normal temperature. It seemed that the GWAS under 22 degree also detected a clear peak at the similar physical position (Fig 1B), indicating the natural variation in PRX62 may also play roles in root hair growth, although this signal did not surpass the significance cutoff. But as the author mentioned that the phenotypic variation under 22 degree was lower than that under 4 degree, which may weaken the detection power. I am curious in Fig 1A, the length of RH difference of the five ecotypes if they continuously grown under 22 degree. Moreover, overexpression PRX62 and PRX69 clearly enhanced RH growth under 22 degree (Fig 3E), and the apoROS level was significantly affected in the double mutant and two overexpressors under 22 degree (Fig 5B). Together, it indicated that alterations in PRX62 (or PRX69) expression level may affect the RH growth under normal temperature. And actually, throughout the manuscript the authors thought that the natural variations in PRX62 gene promoter may change the gene expression. So that I think it can not reach the conclusion that PRX62 only function under low temperature, but it may also play roles in normal temperature in RH growth regulation. I think the author should consider this point.

The authors removed the results regarding that RSL4 transcription factor regulates the PRX69 expression due to the unclear data so far, but appended the insilico analysis of the binding of PRX62 and PRX69 with EXT peptides, and the effect of SHAM chemical treatment to suggest other PRXs may also function in RH growth. It makes the story more focused but less informative.

Overall, this review agreed the work is generally well-done and quality of the revised the version is improved.

Reviewer #2 (Remarks to the Author):

Authors well addressed my comments and I am satisfied with revised MS.

Reviewer #4 (Remarks to the Author):

Overall, the new version of the manuscript has been quite improved and has gained in clarity.

As pointed out by one reviewer, the part with the transcription factor rsl4 was a bit confusing. After removing these results, the story is more flowing. Nevertheless, the end of the introduction needs to be slightly rewritten for mentioning rsl4 is probably not necessary anymore.

The additional details provided about peroxidases and the justification of why prx69 was investigated are very welcome and also participate to the gain in clarity.

The clarification about the statistical analyses and experimental procedures are very appreciated and, I have to say, are even beyond expectations.

The new experiments incorporated in the manuscript well support the previous conclusions and are a big plus. In particular, the additional experiments on extensins are quite of interest. Nevertheless, I think the necessity of correct extensin O-glycosylation for the cross-linking (as predicted by Velasquez et al (cited in manuscript) and shown in vitro by Chen et al. 2015) should be more emphasized. This may prevent from any confusion and would clearly state that even though interaction with PRX62 and PRX69 would happen in extensin regions that are poorly or non-glycosylated, extensins still require to be correctly O-glycosylated in other regions for their cross-linking.

I thank the authors for providing the plots of levels of expression of the enzymes involved in extensin glycosylation, very interesting. The overall upregulation at 10°C is not so spectacular (except for the P4Hs) but may correlate with the observed increased level of extensin.

Some mistakes are still remaining. For example:

Line 210: reveling -> revealing

Line 231 (Jemmat et al. 2020) needs to be formatted

Lines 272 and 276: Figure 4A instead of Figure 4A-B and Figure 4B instead of Figure 4C.

Figure 4 still contains only A and B, no C.

Line 896 have -> has

REVIEWER COMMENTS

Reviewer #1 (Remarks to the Author):

The authors responded to my concerns and revised the manuscript carefully.

I agree with that PRX62 and PRX69 play roles in root hair (RH) growth at low-temperature, however, I am wondering that they function in RH growth under normal temperature. It seemed that the GWAS under 22 degree also detected a clear peak at the similar physical position (Fig 1B), indicating the natural variation in PRX62 may also play roles in root hair growth, although this signal did not surpass the significance cutoff. But as the author mentioned that the phenotypic variation under 22 degree was lower than that under 4 degree, which may weaken the detection power. I am curious in Fig 1A, the length of RH difference of the five ecotypes if they continuously grown under 22 degree. Moreover, overexpression PRX62 and PRX69 clearly enhanced RH growth under 22 degree (Fig 3E), and the apoROS level was significantly affected in the double mutant and two overexpressors under 22 degree (Fig 5B). Together, it indicated that alterations in PRX62 (or PRX69) expression level may affect the RH growth under normal temperature. And actually, throughout the manuscript the authors thought that the natural variations in PRX62 gene promoter may change the gene expression. So that I think it can not reach the conclusion that PRX62 only function under low temperature, but it may also play roles in normal temperature in RH growth regulation. I think the author should consider this point.

R/ Regarding the roles of PRX62 and PRX69 in RH growth at room temperature, the conclusion is still not clear. Based on the lack of RH phenotype for the double mutant *prx62 prx69*, low level of expression specially for PRX62 at room temperature and the GWAS negative result when analyzed at room temperature, all together these suggest that both PRXs they do not play a major role under this condition. We do not discard a minor role for them. At room temperature, we have identify PRX01, PRX44 and PRX73 to play an important role and this is part of another paper published as Preprint and it is currently under revision (Marzol et al. 2020, bioRxiv 2020.02.04.932376).

The authors removed the results regarding that RSL4 transcription factor regulates the PRX69 expression due to the unclear data so far, but appended the insilico analysis of the binding of PRX62 and PRX69 with EXT peptides, and the effect of SHAM chemical treatment to suggest other PRXs may also function in RH growth. It makes the story more focused but less informative.

Overall, this review agreed the work is generally well-done and quality of the revised the version is improved.

R/ We thank the Reviewer for his/her positive appreciation of our work.

Reviewer #2 (Remarks to the Author):

Authors well addressed my comments and I am satisfied with revised MS.

R/ We thank the Reviewer for his/her positive appreciation of our work.

Reviewer #4 (Remarks to the Author):

Overall, the new version of the manuscript has been quite improved and has gained in clarity.

As pointed out by one reviewer, the part with the transcription factor rsl4 was a bit confusing. After removing these results, the story is more flowing. Nevertheless, the end of the introduction needs to be slightly rewritten for mentioning rsl4 is probably not necessary anymore.

The additional details provided about peroxidases and the justification of why prx69 was investigated are very welcome and also participate to the gain in clarity.

The clarification about the statistical analyses and experimental procedures are very appreciated and, I have to say, are even beyond expectations.

The new experiments incorporated in the manuscript well support the previous conclusions and are a big plus. In particular, the additional experiments on extensins are quite of interest. Nevertheless, I think the necessity of correct extensin O-glycosylation for the cross-linking (as predicted by Velasquez et al (cited in manuscript) and shown in vitro by Chen et al. 2015) should be more emphasized. This may prevent from any confusion and would clearly state that even though interaction with PRX62 and PRX69 would happen in extensin regions that are poorly or non-glycosylated, extensins still require to be correctly O-glycosylated in other regions for their cross-linking.

R/ A sentence was added to make this point clearer.

I thank the authors for providing the plots of levels of expression of the enzymes involved in extensin glycosylation, very interesting. The overall upregulation at 10°C is not so spectacular (except for the P4Hs) but may correlate with the observed increased level of extensin.

Some mistakes are still remaining. For example:

Line 210: reveling -> revealing

Line 231 (Jemmat et al. 2020) needs to be formatted

Lines 272 and 276: Figure 4A instead of Figure 4A-B and Figure 4B instead of Figure 4C.

Figure 4 still contains only A and B, no C.

Line 896 have -> has

R/ The manuscript was carefully proofread. We thank this reviewer for all the suggestions.